# A novel rhesus macaque model of Huntington's disease recapitulates key neuropathological changes along with motor and cognitive decline

**Alison R Weiss[1], William A Liguore[1], Kristin Brandon[1], Xiaojie Wang[1,2], Zheng Liu[1,2], Jacqueline S Domire[1], Dana Button[1], Sathya Srinivasan[3], Christopher D Kroenke[1,2,4], Jodi L McBride[1,4]***

[1]Division of Neuroscience, Oregon National Primate Research Center, Beaverton, United States; [2]Advanced Imaging Research Center, Oregon Health and Science University, Portland, United States; [3]Imaging and Morphology Support Core, Oregon National Primate Research Center, Beaverton, United States; [4]Department of Behavioral Neuroscience, Oregon Health and Science University, Portland, United States

**Abstract** We created a new nonhuman primate model of the genetic neurodegenerative disorder Huntington's disease (HD) by injecting a mixture of recombinant adeno-associated viral vectors, serotypes AAV2 and AAV2.retro, each expressing a fragment of human mutant *HTT* (*mHTT*) into the caudate and putamen of adult rhesus macaques. This modeling strategy results in expression of mutant huntingtin protein (mHTT) and aggregate formation in the injected brain regions, as well as dozens of other cortical and subcortical brain regions affected in human HD patients. We queried the disruption of cortico-basal ganglia circuitry for 30 months post-surgery using a variety of behavioral and imaging readouts. Compared to controls, mHTT-treated macaques developed working memory decline and progressive motor impairment. Multimodal imaging revealed circuit-wide white and gray matter degenerative processes in several key brain regions affected in HD. Taken together, we have developed a novel macaque model of HD that may be used to develop disease biomarkers and screen promising therapeutics.

*For correspondence:
mcbridej@ohsu.edu

**Competing interest:** The authors declare that no competing interests exist.

## Editor's evaluation

The authors show the utility of an AAV-based approach in non-human primates to develop an improved model of Huntington's disease. They have presented a convincing, carefully executed, body of work that will be of benefit to a range of researchers studying HD or developing therapies for HD. While this extends the work from an earlier paper (that presented the tools used to induce phenotypes) the results presented are new, relevant, and important to the community.

## Introduction

Huntington's disease (HD) is a genetic, progressive neurodegenerative disorder caused by an expanded CAG/CAA repeat in exon 1 of the *HTT* gene (*Sapp et al., 2001*). When the CAG stretch exceeds approximately 40 repeats, the encoded HTT protein misfolds and sets off a toxic sequence of events inside the cell including transcriptional dysregulation, mitochondrial dysfunction, calcium signaling disruption, and altered neurotransmission that includes glutamatergic and dopaminergic

dysregulation (*Vonsattel et al., 1985*). Accordingly, postmortem studies of HD patient brain tissues have shown significant neuropathology including mutant huntingtin protein (mHTT) aggregate formation, neuronal death, and gliosis (*Sapp et al., 2001*; *Vonsattel et al., 1985*; *Waldvogel et al., 2015*). The most highly affected regions are the caudate nucleus and putamen (together comprising the striatum); however, numerous brain regions that send afferent connections to the striatum are also affected, including widespread cortical and subcortical brain areas (*Rosas et al., 2008*; *Allen et al., 2009*; *Glass et al., 2000*; *Glass et al., 1993*; *Majid et al., 2011*). White matter (WM) fiber tracts interconnecting the striatum with the cortex also show extensive evidence of degeneration (*Sprengelmeyer et al., 2014*; *Stoffers et al., 2010*; *Gregory et al., 2015*; *Saba et al., 2017*), highlighting that HD is not only a striatal disorder, but rather a disease characterized by widespread cortico-basal ganglia involvement.

The symptoms of HD are extremely detrimental to quality of life, as patients suffer from a progressive movement disorder that is characterized by lack of balance and coordination, altered fine motor skills and hyperkinetic involuntary movements of the limbs, trunk, and face, known as chorea. In later stages of the disease patients can experience bradykinetic movements, dystonia, and rigidity (*Mahant et al., 2003*; *Bates et al., 2015*; *Novak and Tabrizi, 2011*). These motor phenotypes are often accompanied by deteriorating cognitive function, including working memory decline and a reduced capacity to plan and organize daily tasks. HD patients also exhibit profound personality changes and mood disturbances that can include depression, anxiety, and irritability (*Tabrizi et al., 2009*; *Lawrence et al., 1996*; *Van den Stock et al., 2015*; *Dogan et al., 2014*; *Julayanont et al., 2020*). Currently, there are no treatments capable of slowing disease progression. Therefore, to aid in the creation and evaluation of effective therapeutic interventions, it is critical to develop appropriate animal models of HD.

Genetically modified rodents have been key in advancing our understanding of pathophysiological mechanisms that are altered by the *mHTT* gene, and numerous mouse and rat models have been created over the past three decades, reviewed in *Farshim and Bates, 2018*. These models have played a critical role in characterizing HTT-mediated disease pathology, and several have been utilized to screen therapeutic strategies. However, an important consideration is that these models do not fully recapitulate the behavioral symptomology of humans with HD. Some HD mouse models display balance issues that can be measured via rotorod performance (*Crook and Housman, 2011*), but do not display the hallmark signs of chorea, bradykinesia nor fine motor skill deficits. Similarly, some mouse models exhibit spatial memory deficits but lack other signature features of HD (*Crook and Housman, 2011*). Furthermore, species differences in genetics, brain size, structure, and neural connectivity all limit the ability to translate findings from rodents to predict clinical responses in human patients. For example, there still exists an ambiguity on whether specific frontal cortical areas in monkeys and humans, such as the medial and lateral prefrontal cortices (areas known to be affected in HD) share cross-species homologies with rodents (*Laubach et al., 2018*). Perhaps most striking to consider is that none of the therapies that have shown efficacy in mouse models have resulted in success in clinical trials over the past several decades. For these reasons, large animal models including sheep, minipigs, and nonhuman primates (NHPs) have emerged more recently as a new and critical tool for the HD research community (*Howland et al., 2020*).

The earliest NHP models of HD were created by using neurotoxins to lesion the caudate and/or putamen in order to replicate the profound striatal atrophy and hyperkinetic movements seen in patients (*Howland et al., 2020*). However, the toxin-based approach decreased in popularity after the *HTT* gene mutation was identified. Subsequent NHP models have been created by viral-mediated delivery of a fragment of the human mHTT gene (*Palfi et al., 2007*; *Weiss et al., 2020*; *Maxan et al., 2020*) or via the development of transgenic HD macaques (*Yang et al., 2008*), both bearing *HTT* genes with expanded CAG repeats that encode mHTT proteins with elongated polyglutamine tracts (Q) at the N-terminus. Palfi et al. created the first viral-mediated NHP model of HD by injecting a lentiviral vector expressing a fragment of *HTT* with 82 CAG repeats (LV-HTT82Q) bilaterally into the dorsolateral posterior portion of the putamen in rhesus macaques (*Palfi et al., 2007*). LV-HTT82Q macaques developed dyskinetic movements of the limbs and trunk beginning at 15 weeks postsurgery and persisting out to 30 weeks post-surgery. Brain tissue revealed mHTT inclusion formation, neuronal loss, and astrocytosis in the circumscribed area of injection. More recently, *Maxan et al., 2020* demonstrated mHTT seeding in the brain (spread of aggregates to nontransduced, neighboring cells), following circumscribed delivery of AAV6-HTT103Q-GFP into the posterior putamen of

adult macaques (). Given that HD pathology in patients has been identified throughout the putamen, caudate nucleus, and in multiple other cortical and subcortical brain regions, it would be of value to create a virally mediated NHP model that more closely recapitulates this widespread pattern of neuro-degeneration and circuit dysfunction. Moreover, because cognitive dysfunction dramatically alters the HD patient's quality of life and autonomy, an NHP model that exhibits both motor and working memory decline will be more useful for evaluating promising therapeutics.

Recent techniques using directed evolution approaches to develop new adeno-associated viruses (AAV) have produced novel capsid variants that exhibit enhanced properties of retrograde and/or anterograde transport in brains of both rodents and NHPs (*Weiss et al., 2020*; *Naidoo et al., 2018*; *Lin et al., 2020*; *Tervo et al., 2016*). Utilizing these new AAV capsid variants, it is now possible to transduce entire neural circuits, rather than individual brain regions. We recently demonstrated that delivery of AAV2.retro expressing a fragment of *HTT* with 85 CAG repeats (AAV2.retro-mHTT85Q) into the adult macaque caudate and putamen results in efficient retrograde transport resulting in mHTT expression and the formation of hallmark aggregates throughout dozens of cortical and subcortical striatal afferents (10 weeks post-surgery). In comparison, injection of the parent serotype, AAV2, expressing mHTT85Q resulted in mHTT inclusions primarily within the caudate and putamen, without appreciable retrograde transport (*Weiss et al., 2020*). This study laid the groundwork for generating a macaque model of HD that engages the entire cortico-basal ganglia circuit using this new viral vector-mediated approach.

Here, we expanded on these initial efforts and characterized the long-term effects of mHTT85Q expression throughout the macaque cortex and basal ganglia. Following surgery, we assessed cognitive and motor function in a cohort of HTT85Q-treated animals, as well as HTT10Q- and buffer-injected controls, for a period of 30 months. Additionally, we performed multimodal imaging to assess WM microstructure, regional brain atrophy and patterns of brain-wide functional connectivity. Together,

**Table 1.** Summary of study participants and surgical cases.

Abbreviations: 85Q, fragment of mHTT protein bearing 85 glutamine repeats; 10Q, fragment of mHTT protein bearing 10 glutamine repeats; Buffer, phosphate buffered saline; VG, vector genomes; SDR, spatial delay response.

| Animal Id | Treatment group | Sex | Age at surgery (years) | Weight at surgery (kg) | Vector dose per hemisphere (vg) | Met criteria for SDR task | Met criteria for lifesaver task |
|---|---|---|---|---|---|---|---|
| 1 | 85Q | F | 10.1 | 5.7 | 3.30E+11 | Yes | Yes |
| 2 | 85Q | F | 11.8 | 6 | 3.30E+11 | Yes | Yes |
| 3 | 10Q | F | 13.5 | 9.9 | 3.30E+11 | Yes | Yes |
| 4 | 85Q | M | 5.9 | 10.7 | 3.30E+11 | Yes | Yes |
| 5 | 10Q | F | 8.7 | 6.5 | 3.30E+11 | Yes | Yes |
| 6 | Buffer | F | 12 | 8.7 | – | Yes | Yes |
| 7 | 10Q | F | 9 | 8.6 | 3.30E+11 | Yes | Yes |
| 8 | Buffer | F | 9.9 | 7.5 | – | No | Yes |
| 9 | Buffer | M | 7.7 | 11.3 | – | Yes | Yes |
| 10 | 10Q | M | 7.5 | 8.8 | 3.30E+11 | Yes | Yes |
| 11 | 85Q | F | 13 | 6.2 | 3.30E+11 | Yes | Yes |
| 12 | 85Q | F | 12.3 | 7.5 | 3.30E+11 | Yes | Yes |
| 13 | Buffer | F | 7.9 | 6 | – | Yes | Yes |
| 14 | 85Q | M | 7.1 | 11.1 | 3.30E+11 | Yes | Yes |
| 15 | Buffer | F | 8.8 | 5.9 | – | Yes | Yes |
| 16 | 10Q | F | 11.8 | 6 | 3.30E+11 | Yes | Yes |
| 17 | 10Q | M | 10 | 9.5 | 3.30E+11 | No | Yes |

this study describes a new macaque model that replicates several of the behavioral and neuropathological changes seen in the early stages of HD.

## Results

To elicit mHTT85Q expression throughout the caudate and putamen, as well as cortical- and basal ganglia afferents, we undertook a combinatorial approach and administered a 1:1 mixture of AAV2-HTT85Q and AAV2.retro-HTT85Q at a titer of 1e12 vg/ml (Group 85Q, $n = 6$). Control animals were injected with a 1:1 mixture of AAV2:AAV2retro-HTT10Q (control HTT fragment with 10 CAG repeats; Group 10Q, $n = 6$) or buffered saline only (Group Buffer, $n = 5$), see *Table 1*. Animals completed a longitudinal battery of cognitive and motor tasks and underwent repeated brain scans to collect structural (T1w/T2w) magnetic resonance imaging (MRI), diffusion tensor imaging (DTI), and resting-state fMRI (rs-fMRI) data.

### HTT85Q administration leads to spatial working memory deficits

Disease onset in HD has historically been defined by the presence of disordered hyperkinetic movements, however HD patients also often experience cognitive decline characterized by working memory impairment, which sometimes precedes motor dysfunction (*Tabrizi et al., 2009*; *Lawrence et al., 1996*; *Julayanont et al., 2020*; *Misiura et al., 2017*; *Paulsen et al., 2001*). In NHPs, spatial working memory can be assessed using a 3-Choice Spatial Delayed Response (SDR) task. Monkeys in this study completed this task at baseline as well as 3, 6, 9, 14, 20, and 30 months post-surgery.

Prior to surgery, 15 animals met the pretraining criteria for the SDR task (*Table 1*), and therefore data from the two remaining subjects were excluded from analysis. For the SDR assay, animals were tasked at remembering the location of a treat-bated, recessed well after a delay of 1, 3, or 5 s (*Figure 1A*). A three-way analysis of variance (ANOVA) found significant main effects of group ($F_{(2,11)} = 17.109$, $p = 0.0004$) and timepoint ($F_{(5,55)} = 2.978$, $p = 0.019$), with no main effect of delay ($F_{(2,22)} = 0.038$, $p = 0.9630$) nor interaction between any of these factors. Post hoc tests revealed that 85Q-treated monkeys performed significantly worse on the SDR task (fewer correct responses) than Buffer- ($p = 0.00014$) and 10Q- ($p = 0.0028$) treated monkeys (*Figure 1B*, delay and timepoint collapsed). Buffer- and 10Q-treated animal performance on the SDR task did not differ significantly ($p = 0.06$). To further explore the time-course of the group differences, we conducted planned comparisons between groups at each timepoint (*Supplementary file 1*). The results indicate that performance between 85Q- and Buffer-treated animals differed significantly at all timepoints. Significant group differences also emerged between 85Q- and 10Q-treated animals at the 6-month timepoint, which persisted through 30 months, with 85Q animals showing working memory deficits that were not exhibited by either control group. Buffer and 10Q groups differed significantly from one another at the 6- and 14-month timepoints, all $p < 0.05$, but both control groups exhibited improvement in their performance over the course of the study (*Figure 1C*, *Supplementary file 1*).

### HTT85Q administration leads to impaired fine motor skill performance

Presymptomatic HD individuals estimated to be 10+ years from disease onset are able to perform self-directed motor tasks (reaching/tapping) similar to age-matched controls, but abnormalities emerge in prodromal HD individuals closer (<5 years) to motor diagnosis and persist for many years (*Misiura et al., 2017*; *Paulsen et al., 2001*; *Stout et al., 2011*). To assess fine motor coordination and skill learning, animals completed a Lifesaver Retrieval Task at baseline as well as 3, 6, 9, 14, 20, and 30 months post-surgery. Prior to surgery, all animals met the pretraining criteria for the Lifesaver Retrieval Task and data from every subject were included in the analysis (*Table 1*). Treat retrieval latency from a baited metal post was measured separately for each hand (*Figure 1D*). A three-way ANOVA with repeated measures for the second and third factors (Timepoint, Hand) revealed a significant main effect of Group ($F_{(2,13)} = 7.892$, $p = 0.006$) and Timepoint ($F_{(5,65)} = 15.953$, $p = 3.121E{-}10$) on changes in Lifesaver retrieval latencies. The main effect of Hand was not significant ($F_{(1,13)} = 1.029$, $p = 0.329$), and there were no significant interactions between this or any other factor. Therefore, data were subsequently collapsed across hands for subsequent analyses.

Compared to Buffer-treated controls, who became quicker on the task compared to their baseline performance, 85Q-treated animals exhibited significantly reduced practice effects in their lifesaver

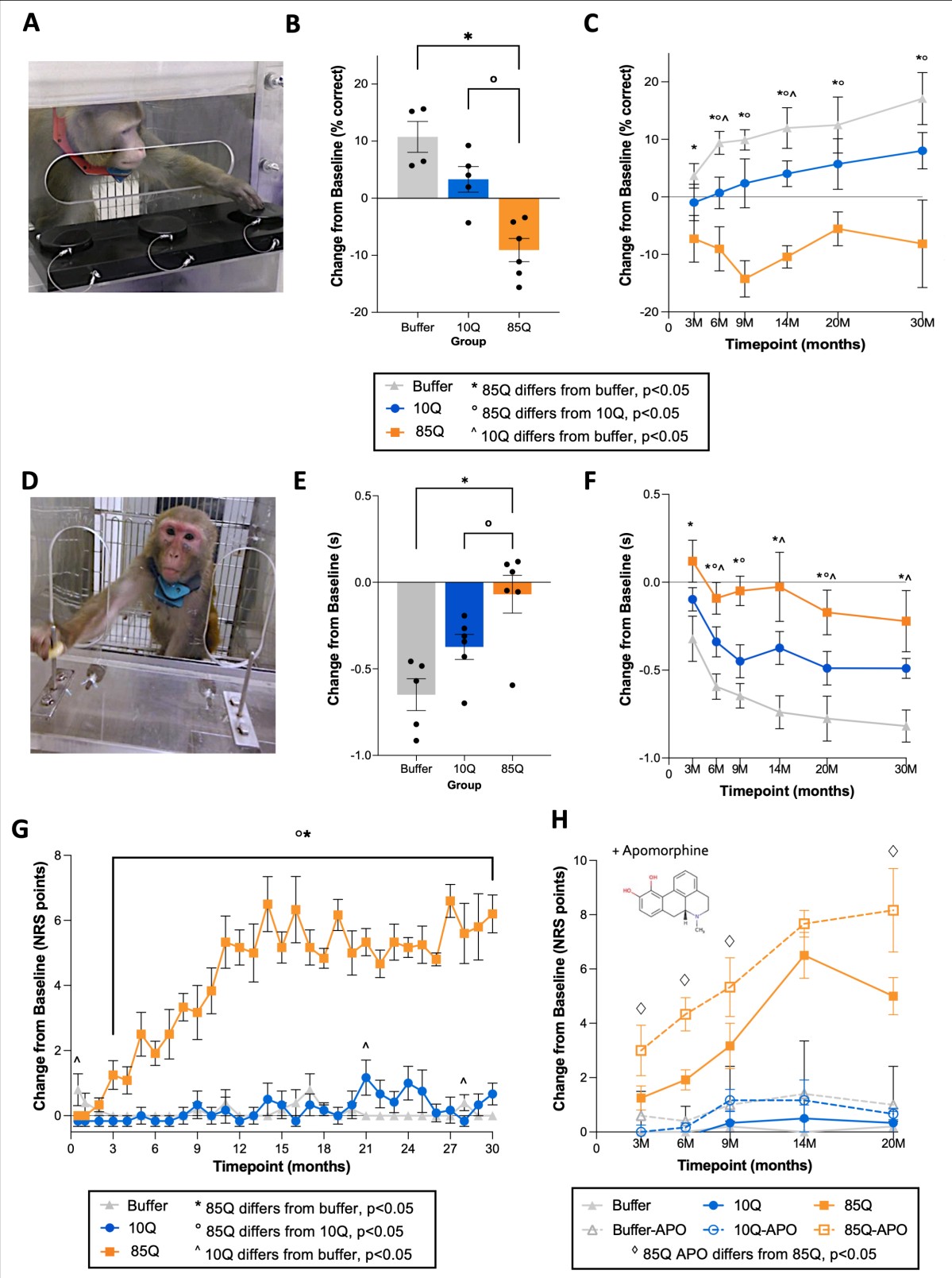

**Figure 1.** Working memory and motor deficits in 85Q-treated animals. (**A**) Example of an animal performing the 3-Choice Spatial Delayed Response (SDR) task. (**B**) Change in SDR performance (% correct) on the SDR task, collapsed across timepoints. (**C**) Change in SDR performance (% correct), expanded across timepoints. All SDR data are expressed as mean ± standard error of the mean (SEM) (85Q – $n = 6$, 10Q – $n = 5$, Buffer – $n = 4$). The key details significant group differences at each timepoint (*, ^, and ° symbols), demonstrating that working memory changes emerge in this model

*Figure 1 continued on next page*

*Figure 1 continued*

beginning at 3–6 months post-85Q administration. (**D**) Example of an animal performing the Lifesaver Retrieval Task. (**E**) Performance collapsed across timepoint, plotted as the change in Retrieval Latency (seconds) from baseline. (**F**) Performance expanded across timepoints, plotted as the change in Retrieval Latency (seconds) from baseline. All Lifesaver Retrieval data are expressed as mean ± SEM (85Q – $n$ = 6, 10Q – $n$ = 6, Buffer – $n$ = 5). The key details significant group differences at each timepoint and shows that fine motor skill learning impairment emerges in 85Q animals as early as 3 months post-85Q administration. (**G**) Mean changes in motor phenotypes plotted for each group. The key shows significant group differences at each timepoint, indicating that motor phenotypes first emerge around 2–3 months post-surgery. (**H**) Neurological rating scale (NRS) scores (difference from baseline) pre- and post-apomorphine administration plotted for each group, with pre-apomorphine NRS scores depicted by solid lines and post-apomorphine NRS scores depicted by dashed lines. The • symbol indicates significant paired comparisons at each timepoint, showing that 85Q-treated animals displayed NRS scores that were modulated by apomorphine at 3, 6, 9, and 20 months post-surgery. All NRS data are expressed as mean ± SEM (85Q – $n$ = 6, 10Q – $n$ = 6, Buffer – $n$ = 5). *p < 0.05, 85Q versus Buffer, °p < 0.05, 85Q versus 10Q, ^p < 0.05, Buffer versus 10Q by three-way analysis of variance (ANOVA).

The online version of this article includes the following source data for figure 1:

**Source data 1.** Spatial Delayed Response (SDR), Lifesaver, and neurological rating scale (NRS) scores.

retrieval latencies, and three animals showed increased latencies (p = 0.002, *Figure 1E*, data collapsed across timepoints). 85Q-treated animals also performed worse on the Lifesaver task compared to 10Q controls (p = 0.049), and performance between Buffer- and 10Q-treated controls did not differ significantly (p = 0.069). To further explore these effects, we conducted planned group comparisons at each timepoint separately (*Supplementary file 2*). The results indicated that 85Q-treated animals differed significantly from Buffer controls at all of the timepoints, and 85Q animals and 10Q controls differed significantly at the 6-, 9-, and 20-month timepoints. In addition, Buffer and 10Q differed significantly at the 6-, 14-, 20-, and 30-month timepoints, all p < 0.05 (*Figure 1F* and *Supplementary file 2*).

## HTT85Q leads to progressive motor phenotypes that are exacerbated by dopamine modulation

Disease onset in HD is defined by the emergence of motor phenotypes, which are scored using the Unified Huntington's Disease Rating Scale (UHDRS; *Huntington study group, 1996*). To assess similar phenotypes in this model, we used an NHP-specific neurological rating scale (NRS) that measures many of the same behavioral phenotypes evaluated in HD patients (*Supplementary file 3*). Similar to the UHDRS, NRS scores range from 0 to 3, with a score of 0 indicating normal behavior and a score of 3 indicating severely impaired behavior, for each phenotype. Monkeys were rated by trained research staff, who were blind to group designation, in their homecage at baseline (prior to surgery), 2 weeks post-surgery and then monthly thereafter for 30 months. Scores for each phenotype category were summed, resulting in a composite score for each timepoint. A two-way ANOVA revealed significant effects of Group ($F$(2,13) = 95.902, p < 0.0001) and Timepoint ($F$(30,390) = 10.657, p < 0.0001), as well as a significant Group × Timepoint interaction ($F$(60,390) = 8.642, p < 0.0001) in total NRS scores. Post hoc tests indicated that 85Q-treated monkeys had significantly higher NRS scores than both Buffer- (p < 0.0001) and 10Q- (p < 0.0001) treated animals, and that the two control groups did not differ (p = 0.878). To further characterize the time-course of these effects, we conducted additional planned comparisons between the groups at each timepoint (*Figure 1G*, *Supplementary file 4*). Results revealed that 85Q-treated animals had significant increases in their total NRS scores post-surgery compared to the Buffer group beginning at the 3-month timepoint, and compared to the 10Q group beginning at the 2-month timepoint, that continuing through the 30-month study duration. Buffer and 10Q exhibited similar NRS scores throughout the duration of the experiment, although group differences between them reached significance at three timepoints (0.5, 21, and 28 m). Beginning at 3 months post-surgery, 85Q-treated animals showed intermittent bouts of orofacial dyskinesia that included lateralized tongue twisting, intermittent tongue protrusion, and lateralized, aberrant mouth and jaw movements that were exacerbated while eating biscuits or treats. Intermittent changes in forelimb posture (outward extension of the arms and downward turning of the hands), particularly in the distal segment of the limb, were also noted by the 3-month timepoint. As time progressed, 85Q-treated animals exhibited increasingly higher NRS scores, with the emergence of additional postural abnormalities (trunk/hindlimb), slowed/dropped treat retrieval, infrequent bouts of forelimb chorea and slowed homecage ambulation. By 9 months post-surgery, some 85Q-treated animals also showed intermittent evidence of mild hindlimb bradykinesia and tremor. These motor phenotypes

continued to persist out to 30 months. Aberrant phenotypes of some Buffer and 10Q animals were noted at small number of timepoints and included abnormal hindlimb weight bearing (Buffer), slowed homecage ambulation (10Q), abnormal forelimb posture (10Q), forelimb tremor (10Q), and orofacial chorea (10Q).

Motor phenotypes exhibited by HD patients are modulated by the neurotransmitter, dopamine. Dopamine receptor agonists have been shown to exacerbate abnormal involuntary movements in early HD patients (*Newman et al., 1985*), and current FDA-approved drugs to treat chorea in HD are the vesicular monoamine transporter 2 (VMAT2) inhibitors, Tetrabenazine (*Huntington Study, 2006*) and Deutetrabenazine (*Frank et al., 2016*), which reduce dopamine availability in the synapse. To query dopamine involvement in the motor deficits observed in our model, we assessed the impact of the nonselective dopamine receptor agonist, apomorphine, on motor behavior. Animals were scored using the NRS immediately prior to, as well as directly following, intramuscular apomorphine administration (0.3 mg/kg) at baseline and at 3, 6, 9, 14, and 20 months post-surgery. NRS scores from 85Q-treated animals were significantly exacerbated by apomorphine, compared to controls. A three-way ANOVA confirmed this observation, revealing significant main effects of Group ($F(2,14) = 62.868$, $p < 0.0001$), Timepoint ($F(4,56) = 11.770$, $p < 0.0001$), and ApoCondition ($F(1,14) = 21.181$, $p < 0.0001$), as well as significant interaction effects between Timepoint and Group ($F(8,56) = 6.021$, $p < 0.0001$), and between ApoCondition and Group ($F(2,14) = 4.503$, $p = 0.031$). To further characterize these effects, we conducted additional planned comparisons between pre- and post-apomorphine scores at each timepoint for each group separately (*Figure 1H*, *Supplementary file 5*). Results indicated that 85Q-treated animals had significantly elevated NRS scores following apomorphine administration, compared to pre-apomorphine administration, at the 3-, 6-, 9-, and 20-month timepoints (all $p < 0.05$), but not at the 14-month timepoint. In contrast, changes in pre- versus post-apomorphine administration scores did not significantly differ at any of the timepoints for Buffer- or 10Q- treated controls.

## mHTT delivery results in microstructural changes in several WM fiber tracts

Microstructural properties of WM fiber tracts can be imaged using DTI, with the most commonly reported metric being fractional anisotropy (FA). Reduced FA is observed in HD in many WM regions including the corpus callosum, superior and inferior longitudinal fasciculi, corona radiata, internal capsule, and cingulum, indicating microstructural changes to the axon including reduced axonal integrity and demyelination (*Sprengelmeyer et al., 2014*; *Stoffers et al., 2010*; *Gregory et al., 2015*; *Saba et al., 2017*). To query WM changes in our model, DTI data were collected pre-surgery and 3, 6, 9, 14, 20, and 30 months post-surgery. Parameter maps of FA, axial diffusivity (AD), radial diffusivity (RD), and mean diffusivity (MD) were generated in ONPRC18 template space (*Weiss et al., 2021*). *Figure 2A* illustrates the results of voxel-wise comparisons of FA, using a WM mask, conducted between the baseline (pre-surgical) timepoint and subsequent timepoints for each group separately. As early as 3 months post-surgery, 85Q-treated monkeys had several regions of WM with significant ($p < 0.01$) reductions in FA that persisted through the 30-month timepoint; in contrast, no changes in FA were observed in 10Q- and Buffer-treated animals.

To characterize the distribution of the significant voxel-wise differences, WM labels from the ONPRC18 macaque brain atlas were applied to the thresholded p-value maps from 85Q-treated animals, and the percent volume of significant FA reduction was calculated for each WM region of interest (ROI) at each timepoint (*Figure 2B*). Broadly, there was an anterior–posterior spatial distribution of WM changes in 85Q animals, with larger relative volumes of reduced FA occurring in more anterior ROIs, without robust hemispheric differences. Of particular note was the large percentages of the corona radiata, dorsal and ventral prefrontal WM tracts, the external and internal capsule, as well as the body and genu of the corpus callosum exhibiting significant reductions in FA compared to baseline. To characterize the overall time-course of these changes, the average change in FA (from baseline) in the significant regions was calculated for each animal at each post-surgical timepoint, collapsed across ROI, and plotted in *Figure 2C*. A repeated measure ANOVA revealed significant main effects of Group ($F(2,13) = 104.117$, $p < 0.0001$) and Timepoint ($F(5,65) = 4.088$, $p = 0.003$), but no interaction between these factors ($F(10,65) = 1.019$, $p = 0.438$). Post hoc tests revealed that 85Q-treated animals had significantly greater decreases in FA after surgery compared to Buffer ($p <$

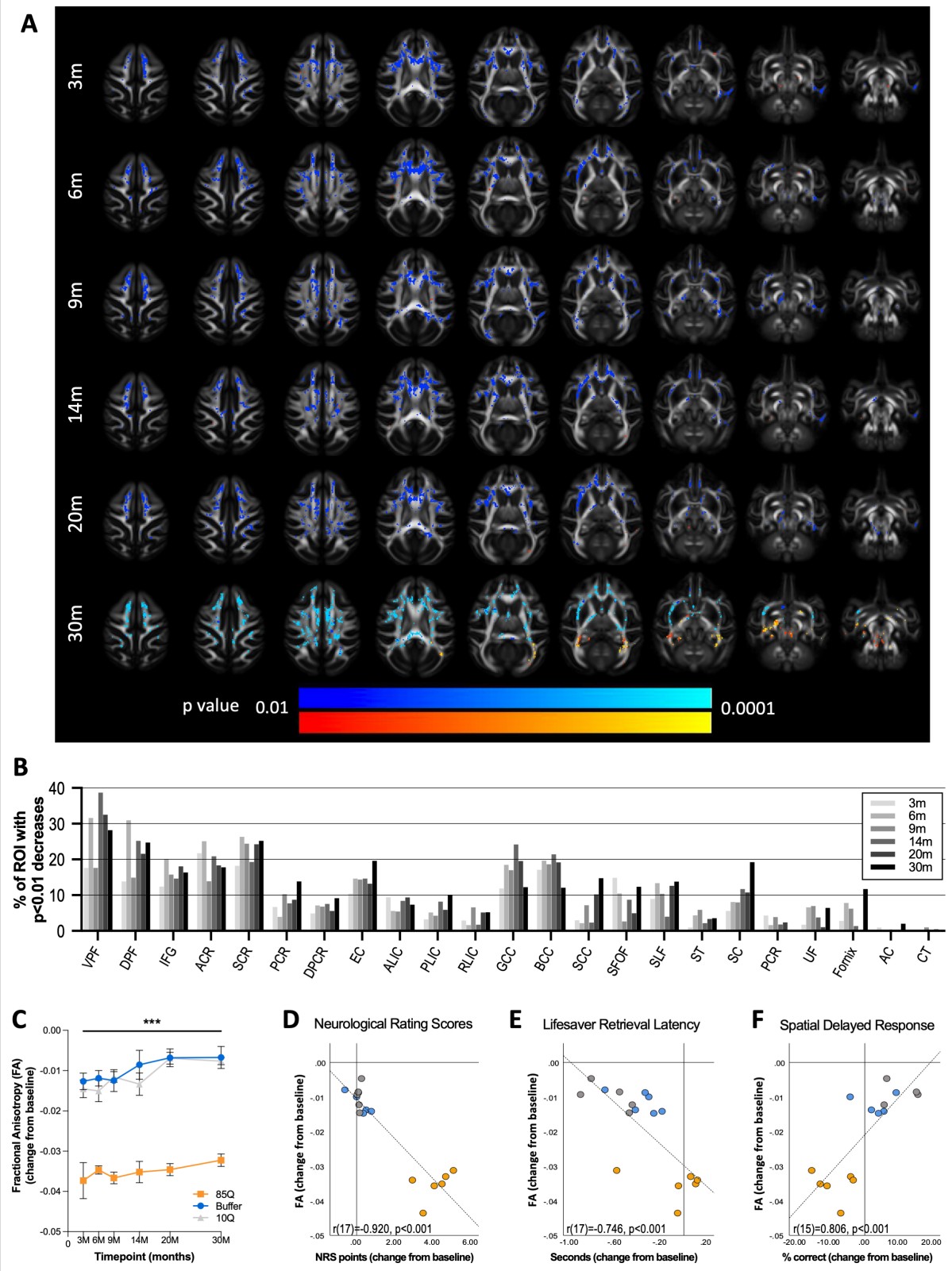

**Figure 2.** 85Q-mediated microstructural changes in cerebral white matter. (**A**) ONPRC18 FA template with overlaying p-value maps shown at a range of p < 0.01 to p < 0.0001. Blue voxels indicate regions of significant FA decrease in Group 85Q, and red voxels indicate regions of significant FA increase. Although there were slight changes in FA in the Buffer- and 10Q-treated animals over time, none of the contrasts reached statistical significance (not pictured). (**B**) Histogram illustrating the percent volume of each ROI where significant FA changes were identified in Group 85Q corresponding to the

*Figure 2 continued on next page*

*Figure 2 continued*

blue voxels in panel (**A**). The line chart in (**C**) illustrates the average magnitude of change in FA under the significant voxels at each timepoint. Data are expressed as mean ± standard error of the mean (SEM). Scatterplots illustrating one-tailed Pearson correlations between FA decreases in white matter (WM) and behavior (both collapsed across timepoint) for three different behavioral measures: (**D**) neurological rating scale (NRS), (**E**) Lifesaver Retrieval Task, and (**F**) Spatial Delayed Response (SDR) task. ***p < 0.001; 85Q differs from Buffer and 10Q. Abbreviations: FA, fractional anisotropy; ROI, region of interest; VPF, ventral prefrontal WM; DPF, dorsal prefrontal WM; IFG, inferior frontal gyrus WM; ACR, anterior corona radiata; SCR, superior corona radiata; PCR, posterior corona radiata; DPCR, dorsal posterior corona radiata; EC, external capsule; ALIC, anterior limb of the internal capsule; PLIC, posterior limb of the internal capsule; RLIC, retrolenticular limb of the internal capsule; GCC, genu of corpus callosum; BCC, body of corpus callosum; SCC, splenium of corpus callosum; SFOF, superior fronto-occipital fasciculus; SLF, superior longitudinal fasciculus; ST, stria terminalus; SC, superior cingulum; PCR, perihippocampal cingulum; UF, uncinate fasciculus; AC, anterior commissure; CT, corticospinal tract.

The online version of this article includes the following source data and figure supplement(s) for figure 2:

**Source data 1.** Diffusion tensor imaging (DTI).

**Figure supplement 1.** 85Q-mediated changes in white matter radial diffusivity (RD).

**Figure supplement 2.** 85Q-mediated changes in white matter mean diffusivity (MD).

**Figure supplement 3.** 85Q-mediated changes in white matter axial diffusivity (AD).

**Figure supplement 4.** 85Q-mediated changes in all diffusivity measures.

0.0001) and 10Q (p < 0.0001) animals, whereas Buffer- and 10Q-treated animals did not differ significantly in the magnitude of FA changes (p = 0.372). To query the relationship between the observed DTI changes and behavioral impairments, we computed correlations between FA changes and behavioral scores that were collapsed across all five post-surgical timepoints. FA decreases in WM were correlated with all three behavioral measures, such that greater decreases in FA were significantly associated with higher NRS scores ($r(17) = -0.920$, p <0.001) (*Figure 2D*), longer Lifesaver retrieval latencies ($r(17) = -0.746$, p < 0.001) (*Figure 2E*) and fewer correct responses on the 3-Choice SDR task ($r(15) = 0.806$, p < 0.001) (*Figure 2F*).

Additional analyses of DTI parameters including RD, MD, and AD were performed, following the same strategy described for FA. 85Q-treated animals showed significant elevations in RD (*Figure 2—figure supplement 1*) and MD (*Figure 2—figure supplement 2*), as well as significant decreases in AD (*Figure 2—figure supplement 3*), compared to controls (p < 0.01 for each measurement). These changes followed similar spatiotemporal trajectories to the changes observed in FA. In particular, there were areas of significant RD and MD increase in corresponding regions of FA decrease in the dorsal and ventral prefrontal WM tracts, as well as areas of the anterior/dorsal corona radiata, internal capsule, external capsule, and the genu and body of the corpus callosum. Brain-wide, the spatial distributions of RD and MD increases generally overlapped the FA decreases, whereas the significant decreases in AD were more widely distributed (*Figure 2—figure supplement 4*). None of the changes in AD, RD, or MD observed in the 10Q and Buffer groups survived thresholding at the p < 0.01 level.

## Cortico-striatal atrophy revealed with tensor-based morphometry

Tensor-based morphometry (TBM) is a technique used to assess changes in brain morphometry by comparing deformation fields that align anatomical MR images to a template image. Applied longitudinally, changes in these deformation fields can be used to characterize 3D patterns of structural change. Atrophy of many brain regions (caudate, putamen, thalamus, multiple regions of cortex) has been described in symptomatic HD patients using this technique (*Hobbs et al., 2010*; *Kassubek et al., 2004*), and recent longitudinal data indicate that some changes begin prior to overt symptom development; particularly in regions of the putamen and globus pallidus (*Stoffers et al., 2010*; *Hobbs et al., 2010*; *Kipps et al., 2005*). High-resolution (0.5 mm isotropic) T2-weighted SPACE scans were collected pre-surgery and 3, 6, 9, 14, 20, and 30 months post-surgery. For each timepoint, deformation fields aligning with the ONPRC18 template were computed, and log Jacobian Determinant maps, reflecting local structural changes, were generated. In the resulting maps, positive values indicated areas where individual space was dilated to align with ONPRC18 space (and was therefore smaller), whereas negative values indicate areas where individual space was contracted to align with ONPRC18 space (and was therefore larger). Using a GM mask, voxel-wise comparisons were conducted to compare the baseline (pre-surgical) timepoint and subsequent timepoints. As early as 3 months post-surgery, 85Q-treated monkeys exhibited regions of the basal ganglia, prefrontal- and

premotor-cortex with significant tissue contractions (larger log Jacobian Determinants), indicating tissue atrophy, compared to baseline that persisted through the 30-month timepoint (p < 0.01). In contrast, the none of the changes observed in 10Q and Buffer survived thresholding at the p < 0.01 level (*Figure 3A*). GM labels available in the ONPRC18 atlas were applied to the thresholded p-value maps from 85Q-treated animals, and the percent volumes of significant log Jacobian Determinant increases were calculated for each ROI at each timepoint. Major brain areas showing tissue contraction were the caudate, putamen, globus pallidus (internal and external segments), as well as numerous frontal and motor cortical areas, with no hemispheric differences (*Figure 3B*).

To characterize the overall time-course of these changes, masks of thresholded p-value maps from each timepoint were created for areas of significant contraction. The change in the log Jacobian Determinant maps (from baseline) under the masks was then calculated for each animal at each post-surgical timepoint and plotted in *Figure 3C*. In the regions with significant increases in log Jacobian Determinants, a two-way repeated measure ANOVA revealed significant main effects of Group ($F(2,13)$ = 19.568, p < 0.001) and Timepoint ($F(5,65)$ = 22.180, p < 0.001), but no interaction between these factors ($F(10,65)$ = 0.938, p = 0.505). Post hoc tests indicated that 85Q-treated animals had significantly greater tissue contractions (greater increases in average log Jacobian Determinant values) compared to Buffer- (p < 0.001) and 10Q- (p < 0.001) treated animals, with no differences between control groups (p = 0.203).

Similar to other imaging measurements, correlations were characterized between the observed TBM results and the behavioral measures; one-tailed Pearson correlations were computed between log Jacobian Determinant changes and behavioral change scores that had been collapsed across all five post-surgical timepoints. In regions with significant tissue atrophy, TBM changes in GM were correlated with all three behavioral measures, such that greater contractions were significantly associated with higher NRS scores ($r(17)$ = 0.786, p < 0.001) (*Figure 3D*), longer Lifesaver retrieval latencies ($r(17)$ = 0.730, p < 0.001) (*Figure 3E*) and fewer correct responses on the 3-Choice SDR task ($r(15)$ = −0.683, p = 0.002; *Figure 3F*).

## Reduced cortico-basal ganglia functional connectivity

rs-fMRI is a technique used to identify areas of the brain that exhibit correlated fluctuations in blood oxygenation level-dependent (BOLD) MR signal intensity in the absence of a specific stimulus (hence, 'resting state'; *Lv et al., 2018*). Independent component analysis (ICA) of rs-fMRI is a data-driven computational method used to decompose correlated patterns of BOLD signals into networks (or 'components') with distinct spatial and temporal characteristics, without the reliance on pre-established user-defined ROIs for seed-based correlations. Using ICA, we identified four independent resting-state components (ICs) of interest using data collected at the baseline timepoint from all of the study animals. IC1 includes areas of the occipital cortex and ventrolateral PFC; IC2 includes the caudate, putamen, ventral sensory-motor cortex, and medial PFC; IC3 includes the caudate and several prefrontal and temporal cortical regions; and IC4 includes the caudate, putamen, cingulate, dorsal-prefrontal, and dorsal-motor areas (*Figure 4—figure supplement 1*). To facilitate longitudinal group-level comparisons, dual-regression (DR) analysis was applied to each timepoint, focusing on these four ICs, and z-score connectivity maps for each of the four ICs from each monkey at each timepoint were subsequently generated.

In the field of HD, studies applying rs-fMRI with ICA-DR (*Dumas et al., 2013*; *Hohenfeld et al., 2018*; *Seibert et al., 2012*; *Werner et al., 2014*) indicate that areas of decreased RSFC emerge first in premanifest HD-gene carriers, which shifts later in disease stages to increased RSFC, potentially as a compensatory mechanism. To define changes in RSFC in our model, we conducted voxel-wise comparisons for each IC between z-score maps from the baseline (pre-surgical) timepoint and subsequent timepoints using a whole brain mask. The network exhibiting the most protracted changes over the study timeline was component IC2, which involves striatal and sensory-motor areas (*Figure 4*, *Figure 4—figure supplement 1*). As early as 3 months post-surgery and continuing through the 30-month timepoint, 85Q-treated monkeys showed significant decreases in IC2 z-scores in many cortical and subcortical areas (p < 0.01). In contrast, none of the changes observed in 10Q and Buffer survived thresholding at the p < 0.01 level (*Figure 4A*). The percent volume of significant change in z-scores with IC2 were calculated for each ROI in the ONPRC18 labelmap at each timepoint and plotted in *Figure 4B*. The results highlighted regions of the VLPFC, VMPFC, SSC, IC as well as areas

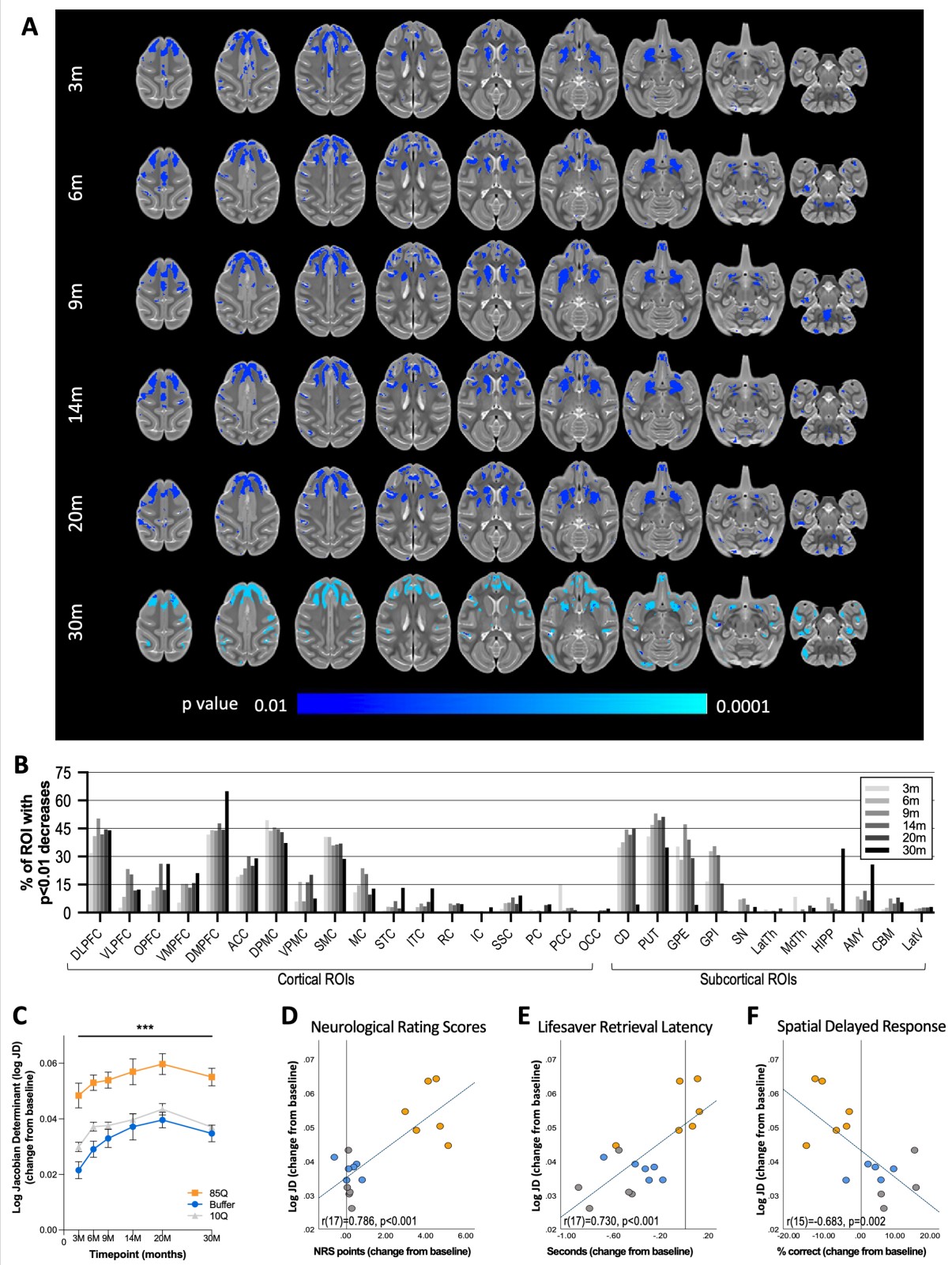

**Figure 3.** 85Q-mediated tissue atrophy in cortico-basal ganglia gray matter. (**A**) ONPRC18 T2w template with overlaying p-value maps shown at a threshold of p < 0.01 to p<0.0001. Blue voxels indicate regions of significant TBM contraction (increased log Jacobian Determinants) in Group 85Q. Although there were slight changes in the Buffer- and 10Q-treated animals over time, none of the contrasts reached statistical significance (not pictured). (**B**) Histogram illustrating the percent volume of each cortical and subcortical ROI where significant TBM contractions were identified

*Figure 3 continued on next page*

*Figure 3 continued*

(corresponding to the blue voxels in A). (**C**) A mask that merged together the thresholded p-value maps from each timepoint was created. Line charts illustrate the average log Jacobian Determinate changes under this mask for each group separately. Data are expressed as mean ± standard error of the mean (SEM) (85Q – $n$ = 6, 10Q – $n$ = 6, Buffer – $n$ = 5), repeated measures analysis of variance (ANOVA). (**D**) Scatterplots illustrating two-tailed Pearson correlations between TBM contractions in GM and behavior (both collapsed across timepoint) for three different behavioral measures: (**D**) neurological rating scale (NRS), (**E**) Lifesaver Retrieval Task, and (**F**) Spatial Delayed Response (SDR) task. ***p < 0.01; 85Q differs from Buffer and 10Q. Abbreviations: TBM, tensor-based morphometry; ROI, region of interest; LogJD, log Jacobian Determinant; DLPFC, dorsolateral prefrontal cortex; VLPFC, ventrolateral prefrontal cortex; OPFC, orbitofrontal cortex; VMPFC, ventromedial prefrontal cortex; DMPFC, dorsomedial prefrontal cortex; ACC, anterior cingulate cortex; DPMC, dorsal premotor cortex; VPMC, ventral premotor cortex; SMC, supplemental motor cortex; MC, primary motor cortex; STC, superior temporal cortex; ITC, inferior temporal cortex; RC, rhinal cortex; IC, insular cortex; SSC, somatosensory cortex; PC, parietal cortex; PCC, posterior cingulate cortex; OCC, occipital cortex; CD, caudate; PUT, putamen; LatTH, lateral thalamus; MdTH, medial thalamus; HIPP, hippocampus; AMY, amygdala; SN, substantia nigra; GPI, internal globus pallidus; GPE, external globus pallidus; CBM, cerebellum; LatV, lateral ventricles.

The online version of this article includes the following source data for figure 3:

**Source data 1.** Tensor-based morphometry (TBM).

of the putamen, substantia nigra, globus pallidus, amygdala, and hippocampus with significantly reduced connectivity (p <0.01) to the IC2 network over the 30-month study timeline.

To characterize the overall time-course of these changes, the change in IC2 *z*-scores (from baseline) in these significant regions were calculated for each animal at each post-surgical timepoint and plotted in *Figure 4C*. A two-way repeated measure ANOVA revealed a significant main effect of Group ($F$(2,13) = 51.108, p < 0.0001), but no effect of Timepoint ($F$(5,65) = 0.666, p = 0.650) nor interaction ($F$(10,65) = 0.735, p = 0.689). Post hoc tests indicated that 85Q animals had significantly larger decreases in IC2 *z*-scores after surgery compared to the Buffer (p < 0.0001), and 10Q (p < 0.0001) animals, with no significant differences in the magnitude of *z*-score changes between controls (p = 0.742).

To probe the relationship between the changes in IC2 connectivity and the behavioral measures, correlations were computed between the *z*-score changes and behavioral change scores, collapsed across all five post-surgical timepoints. Similar to the changes we observed using the other imaging modalities, the magnitude of decreases in RSFC with IC2 correlated with all three behavioral measures such that larger decreases in RSFC were associated with higher NRS scores ($r$(17) = −0.924, p < 0.001) (*Figure 4D*), longer Lifesaver retrieval latencies ($r$(17) = −0.672, p = 0.003) (*Figure 4E*) and lower percent correct on the 3-Choice SDR task ($r$(15) = 0.857, p < 0.001) (*Figure 4F*).

In addition, significant changes emerged in IC3 at the 30-month timepoint, such that 85Q animals exhibited large regions of reduced functional connectivity with this network that were not apparent at the earlier times (*Figure 4—figure supplement 4*). Areas of the prefrontal cortex, caudate, and medial thalamus were particularly impacted. There were also modest, and transient changes in the functional connectivity of IC1, and IC4 over the 30-month study timeline. In 85Q-treated monkeys, areas of the occipital and prefrontal cortex exhibited an initial increase in connectivity with IC1 that returned to baseline levels by 6 months post-surgery (*Figure 4—figure supplement 3*). The changes observed in IC4 were more subtle (*Figure 4—figure supplement 5*), showing only small regions of cortex with altered RSFC compared to baseline. In all cases, however, none of the changes in IC1, IC3, or IC4 observed in 10Q and Buffer survived thresholding at the p < 0.01 level.

## HTT85Q expression leads to aggregate formation

Following the final data collection point at 30 months post-surgery, animals were taken to necropsy and brains were collected for postmortem histological evaluation. Serial tissue sections throughout the brains of all animals were immunohistochemically stained using em48, an antibody that preferentially binds to polyQ-expanded HTT (82–150Q) and has been used widely to detect mHTT protein aggregates in human HD tissue samples as well as tissues from various animal models. We detected a mixture of both small and large em48+ aggregates in numerous brain regions of 85Q-treated macaques, but not in 10Q- or buffer-injected controls. Specifically, em48+ aggregates were found throughout the head of the caudate and throughout the putamen, corresponding to the areas of injection, as well in the deeper layers (V/VI) of several of cortical regions that send afferent projections to these regions, including expression in the dorsal and ventral prefrontal, dorsal and ventral premotor, supplemental motor, anterior cingulate, primary motor, and insular cortices, as well as lower levels of

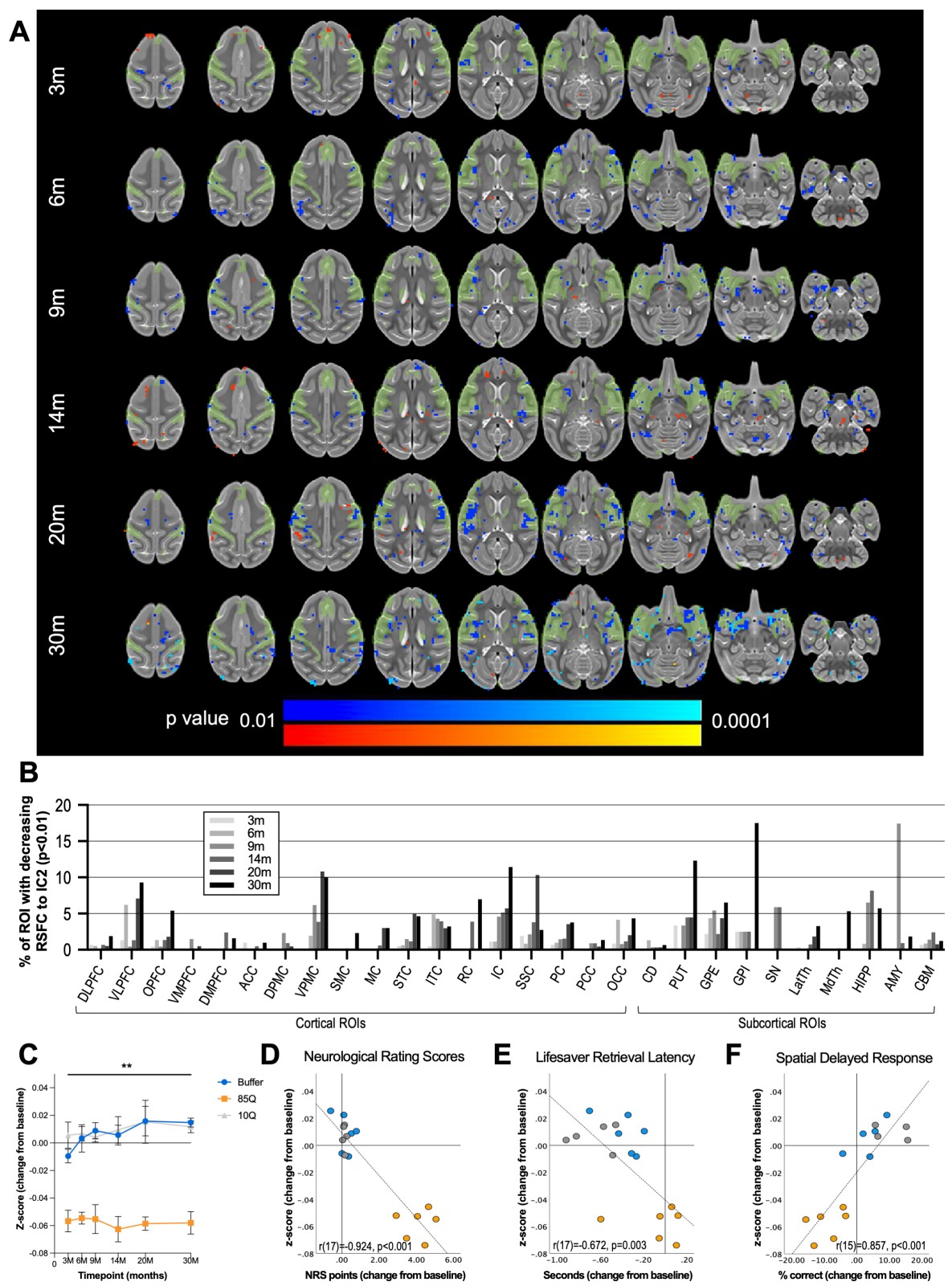

**Figure 4.** 85Q-mediated alterations in patterns of brain-wide resting-state functional connectivity. (**A**) ONPRC18 T2w template with overlaying map of voxels showing high temporal correlation with independent component 2 (IC2), depicted in green. Additional overlaying p-value maps are shown at a threshold of p < 0.01 to p < 0.0001. Blue voxels indicate regions of significantly reduced RSFC (decreased z-score) with IC2 in Group 85Q, and red voxels indicate regions of significant increased RSFC (increased z-score) with IC2. Although there were slight changes in the Buffer- and 10Q-treated

*Figure 4 continued*

animals over time, none of the contrasts reached statistical significance (not pictured). (**B**) Histogram illustrating the percent volume of each cortical and subcortical region of interest (ROI) where significantly decreased *z*-scores were identified at each timepoint (corresponding to the blue voxels in A). (**C**) A mask that merged together the thresholded p-value maps from each timepoint was created for regions of RSFC decreases with IC2. Line charts illustrate the average magnitude of RSFC changes (from baseline) under this mask for each group separately, ±1 standard error of the mean (SEM). (**D**) Scatterplots illustrating two-tailed Pearson correlations between RFSC changes in GM and behavior (both collapsed across timepoint) for three different behavioral measures: (**D**) neurological rating scale (NRS), (**E**) Lifesaver Retrieval Task, and (**F**) Spatial Delayed Response (SDR) task. RSFC, resting-state functional connectivity. **p < 0.001; 85Q differs from Buffer and 10Q. Abbreviations: DLPFC, dorsolateral prefrontal cortex; VLPFC, ventrolateral prefrontal cortex; OPFC, orbitofrontal cortex; VMPFC, ventromedial prefrontal cortex; DMPFC, dorsomedial prefrontal cortex; ACC, anterior cingulate cortex; DPMC, dorsal premotor cortex; VPMC, ventral premotor cortex; SMC, supplemental motor cortex; MC, primary motor cortex; STC, superior temporal cortex; ITC, inferior temporal cortex; RC, rhinal cortex; IC, insular cortex; SSC, somatosensory cortex; PC, parietal cortex; PCC, posterior cingulate cortex; OCC, occipital cortex; CD, caudate; PUT, putamen; GPE, external globus pallidus; GPI, internal globus pallidus; SN, substantia nigra; LatTH, lateral thalamus; MdTH, medial thalamus; HIPP, hippocampus; AMY, amygdala; CBM, cerebellum.

The online version of this article includes the following source data and figure supplement(s) for figure 4:

Source data 1. IC2 RSFC.

Figure supplement 1. Resting-state networks of interest identified using independent component (IC) analysis.

Figure supplement 2. Networks identified using independent component (IC) analysis associated with noise.

Figure supplement 3. 85Q-mediated changes resting-state functional connectivity: IC1.

Figure supplement 4. 85Q-mediated changes resting-state functional connectivity: IC3.

Figure supplement 5. 85Q-mediated changes resting-state functional connectivity: IC4.

expression in the orbitofrontal, somatosensory, temporal and parietal cortices. Subcortical regions that contained em48+ aggregates included the amygdala, thalamus, globus pallidus, claustrum, and to a lesser extent, the hippocampus and substantia nigra. *Figure 5A* shows low (×4) and high (×20) magnification brightfield photomicrographs depicting examples of highly transduced cortical and subcortical brain regions. Triple-label immunofluorescence staining determined that em48+ mHTT aggregates were located in neurons (*Figure 5B*) but not detected in astrocytes (*Figure 5C*) in both the areas of injection (example from the putamen) as well as in distal cortical areas (example from the supplemental motor cortex). Expression was primarily restricted to the nucleus (*Figure 5B, C*; em48/Hoechst coexpression), where both diffuse and aggregated mHTT expression was routinely observed; however, small extracellular em48+ foci could also be observed in the neuropil in several gray matter regions. Additionally, em48+ aggregates were seen in localized areas in some WM tracts, including the corona radiata, corpus callosum, superior cingulum, internal capsule, external capsule, and prefrontal WM tracts (data not shown).

Immunohistochemical staining with the N-terminal HTT antibody, 1–82 aa (2B4 clone), which recognizes both mutant and wildtype HTT, matched the same pattern of nuclear em48 expression seen in 85Q-treated animals, but showed diffuse, cytoplasmic staining in neurons from 10Q-treated control animals in the same brain regions. Buffer-treated animals showed only very light, near background levels of staining throughout the brain, likely owing to HTT sequence homology between humans and macaques from aa 1 to 82. This pattern of staining is similar to what has been reported in human HD and control cases (dark 2B4+ nuclear staining in HD cases and diffuse, lighter cytoplasmic staining in control cases; *Herndon et al., 2009*).

## Discussion

The goal of this study was to develop a macaque model of HD that recapitulates neurodegenerative changes throughout the cortico-basal ganglia network and gives rise to characteristic motor and cognitive decline seen in early-stage human HD patients. Secondarily, we sought to characterize the potential relationships between behavioral phenotypes and neuroimaging findings of disease progression in order to define a set of reliable outcome measures that can be used in future translational studies of candidate HD therapeutics in this model. Because the striatum is the earliest and most severe brain region affected in HD, with degeneration extending to WM and eventually widespread regions of the cortex (*Vonsattel et al., 2011*), we undertook a unique approach delivering a mixture of AAV2 and AAV2.retro in order to express mHTT throughout the entire macaque cortico-striatal circuit.

 Research article

Neuroscience

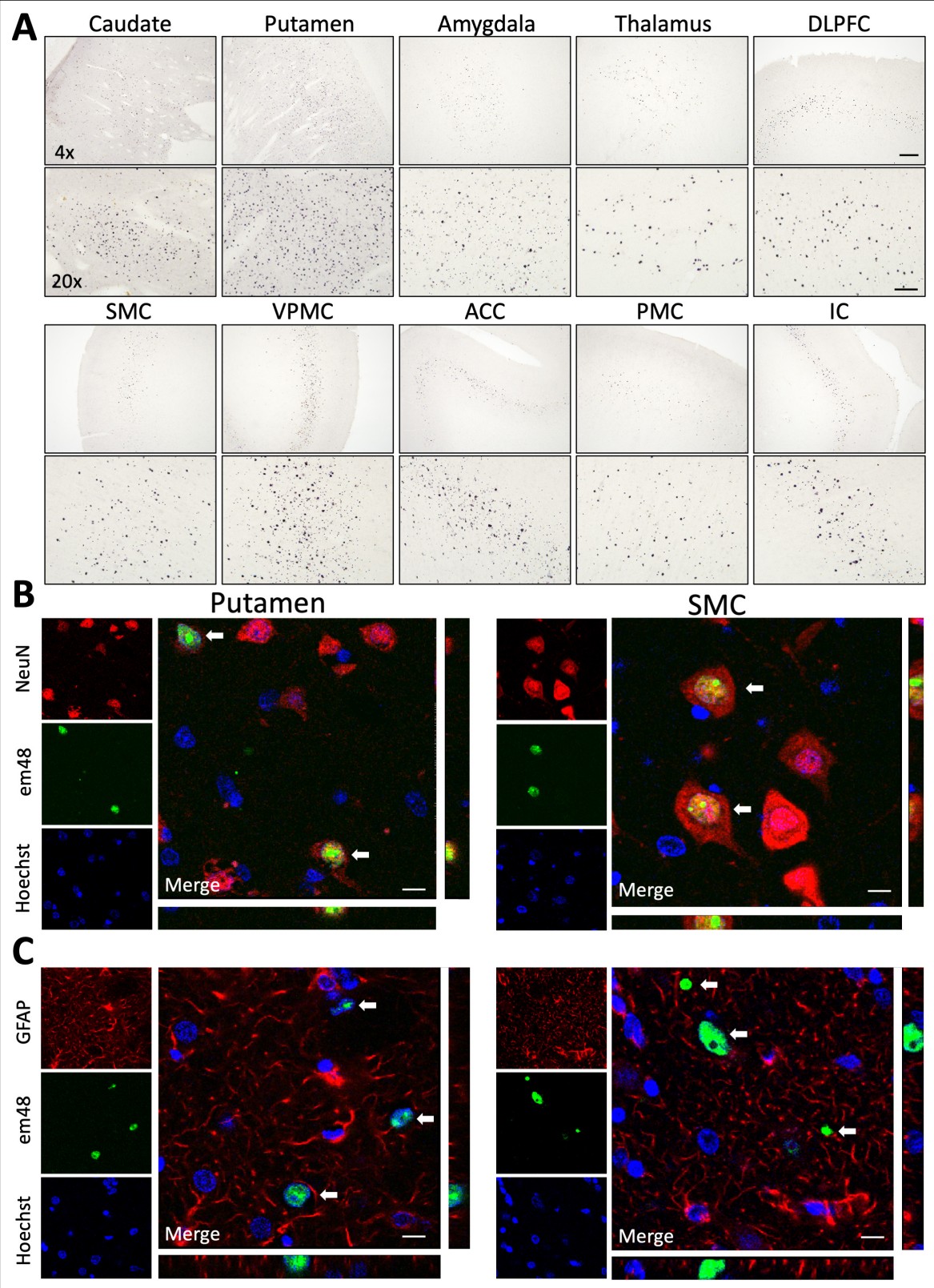

**Figure 5.** 85Q-treated animals show evidence of neuronal em48+ mHTT protein inclusion body formation throughout cortical and subcortical brain regions. (**A**) Low (×4) and high (×20) magnification brightfield photomicrographs illustrate em48+ mHTT inclusions throughout the areas of injection (caudate and putamen), as well as in several other cortical and subcortical brain regions with known afferent projections to the striatum. Examples shown here: DLPFC, dorsolateral prefrontal cortex; SMC, supplemental motor cortex; VPMC, ventral premotor cortex; ACC, anterior cingulate cortex;

*Figure 5 continued on next page*

*Figure 5 continued*

PMC, primary motor cortex; IC, insular cortex. Scale bar on ×4 image indicates 200 µM, scale bar on ×20 image indicates 50 µM. em48+ inclusions were not observed in 10Q- or buffer-treated controls. (**B**) Triple-labeled immunofluorescent confocal images from 85Q-treated animals illustrating localization of mHTT inclusions in neuronal nuclei. Examples shown here are from an area of injection (putamen – left panel) as well as a distal cortical region (supplemental motor cortex – right panel). White arrows indicate cells that are triple labeled for neurons (NeuN+), mHTT inclusions (em48+), and nuclei (stained with Hoechst 33342). (**C**) Triple-labeled immunofluorescent confocal images illustrating the lack of localization of mHTT inclusions in astrocytes, examples shown from the putamen and supplemental cortex. White arrows indicate cells that are labeled for mHTT inclusions (em48+) and nuclei (stained with Hoechst 33342), but not astrocytes (GFAP+). Scale bars in B and C = 10 µM. Orthogonal views from different planes are included for reference, and depict an example of one of the em48+ cells from each brain region, and for each cellular marker.

Over the 2.5-year (30-month) study timeline, HTT85Q animals displayed motor and cognitive deficits that were accompanied by microstructural alterations, atrophy, and reduced functional connectivity throughout the cortico-basal ganglia network. Due to the long scanning times achievable with anesthetized NHPs, high-resolution DTI enabled a more detailed analyses of WM tracts than has been previously achieved. The reduction in WM FA seen in HTT85Q macaques, a biomarker associated with myelin breakdown and axonal swelling, closely mirrors changes seen in human HD patient WM tracts, including the prefrontal WM tracts, corona radiata, internal and external capsules, and the corpus callosum (*Estevez-Fraga et al., 2020*). In addition to microstructural changes detected by DTI, HTT85Q-treated animals also exhibited subtle striatal and cortical atrophy, along with mild reductions in network connectivity, which were present by the first timepoint of analysis (3 months post-AAV-administration). The overall patterns of neurodegeneration that we characterized in this study models similar changes detected in prodromal and early-stage HD patients (*Stoffers et al., 2010*; *Hobbs et al., 2010*; *Kipps et al., 2005*; *Dumas et al., 2013*; *Hohenfeld et al., 2018*; *Seibert et al., 2012*; *Werner et al., 2014*). Future evaluation at even earlier timepoints will help address whether degenerative processes begin in the striatum first, before being detectable in WM and afferent cortical areas – as is seen in human HD cases – or whether these pathological changes occur simultaneously in this AAV-based model.

Beginning at 3 months post-surgery, 85Q-treated macaques developed mild motor phenotypes similar to those experienced by HD patients including incoordination, forelimb and orofacial chorea, hindlimb tremor and postural changes in the distal portion of the forelimbs. Over time, these behaviors worsened in severity, and further changes were observed including evidence of delayed gait initiation and hindlimb bradykinesia during locomotion. Furthermore, the severity of these phenotypes was exacerbated by the dopamine agonist, apomorphine (*Figure 1H*). Apomorphine has a high binding affinity for dopamine $D_2$, $D_3$, and $D_5$ receptors, and altered dopamine neurotransmission has been well documented in HD; for review, *Cepeda et al., 2014*. Positron emission tomography (PET) studies have confirmed a progressive loss of striatal D2 and D1 receptor density throughout the course of disease that correlate with clinical severity assessed using motor UHDRS and Total Functional Capacity (TFC) scales (*Weeks et al., 1996*; *Andrews et al., 1999*). PET studies are planned to characterize potential dopamine receptor density alterations in this model. Overall, the cognitive and motor phenotypes exhibited in HTT85Q macaques model some of the same symptoms experienced by patients in the early stages of their disease development.

Histological evaluation of brain tissue at 30 months post-surgery confirmed the presence of mHTT protein expression and pathological aggregate formation in the brains of 85Q-treated animals compared to controls. em48+ mHTT aggregates were detected in the same cortical and subcortical brain regions throughout the cortico-basal ganglia circuit that we previously characterized in animals assessed at 10 weeks post-administration of AAV2.retro-HTT85Q (*Weiss et al., 2020*). This finding suggests that mHTT aggregation in this macaque model is stable and does not involve additional brain regions over time. These data are in line with evidence from human HD postmortem brain tissues, showing that mHTT aggregate density in most cortical regions does not increase with worsening Vonsattel Grading (i.e., no difference between Grades 1–4), but rather is more dependent on the CAG repeat length, with higher CAG repeats correlating with mHTT aggregate density (*Hickman et al., 2022*). Our data show an anterior to posterior gradient of em48+ expression, with the highest levels of expression in prefrontal and premotor cortices, moderate levels in motor and somatosensory cortices, less expression in temporal and parietal cortices and little to no expression in the occipital cortices. These findings are in concordance with the pattern of aggregate density seen in human HD

cases, with successive decreases in density seen from rostral to caudal cortices (BA9 > BA4 > BA7 > BA17) (*Hickman et al., 2022*). Our observations of em48+ inclusions predominantly in deeper cortical layers (V/VI) in 85Q-treated animals also matches the pattern seen in human HD brain tissue, where HTT+ aggregates predominate in the deeper, infragranular layers of the cortex (V/VI). These cortical layers serve as the primary output layers sending axons to subcortical structures, including the caudate and putamen (*Hickman et al., 2022*).

We confirmed robust levels of em48+ mHTT aggregates 85Q-treated animals in the same brain regions that showed the most significant changes in brain atrophy identified using TBM, including the caudate, putamen, prefrontal, premotor supplemental motor and anterior cingulate cortices. Moreover, em48+ inclusions were detected in the same WM tracts showing significant reduction in FA via DTI, including the ventral and dorsal prefrontal WM tracts, the anterior and superior corona radiata, the genu and body of the corpus callosum and the superior cingulum. em48+ aggregates in WM were largely located in, and in regions surrounding, the needle tracts. To prevent clogging of the needle, the infusion pump was run at a low rate (1 ul/min) while the needle was being lowered into the target brain regions; therefore, a small amount of viral vector was dispersed into WM along the needle tract.

Future histological studies are planned in this model to investigate potential gray matter neuronal loss, WM fiber loss and/or changes in myelin density. These data will be correlated with the imaging, behavioral and histopathological findings reported here. Additionally, given that the HTT fragment used in this model contains 83 pure CAGs, followed by a CAA/CAG cassette (single CAA interruption), the possibility for somatic instability cannot be ruled out and will be investigated using brain tissues collected from this same cohort. If present, we hypothesize that repeat expansion will be the greatest in the caudate and putamen, based on similar findings in human HD cases (*Swami et al., 2009*).

While the mouse models of HD developed to date have clear advantages (e.g., the relative ease of generating transgenic and knock-in animals, and the fact that they model some of the mHTT-mediated genetic changes (e.g., somatic instability) and neuropathological intracellular cascades seen in human HD), they do not replicate the complex array of behavioral symptoms seen in human patients, including chorea. Moreover, their much smaller brain size (~3700-fold smaller by weight) makes scaling up drug delivery strategies to humans an arduous task, particularly with a delivery that requires complex neurosurgery to deep brain structures. Work with large animal models of HD has accelerated greatly over the past decade due to the development of viral-based and genetically modified sheep, minipigs, and NHPs (*Palfi et al., 2007*; *Yang et al., 2008*; *Jacobsen et al., 2010*; *Baxa et al., 2013*). For a comprehensive review, see *Howland et al., 2020*. Transgenic HD sheep do not exhibit many of the overt behavioral phenotypes seen in human HD patients, aside from circadian disturbances, but do show signatures of mHTT-mediated pathology including the development of cortical mHTT-positive inclusions, evidence of increased brain urea, as well as loss of cannabinoid receptor 1 (CB1) and dopamine- and cyclic AMP-regulated phosphoprotein (DARPP-32) in the globus pallidus (*Jacobsen et al., 2010*; *Reid et al., 2013*; *Handley et al., 2016*; *Handley et al., 2017*). While HD sheep do not develop cognitive and motor phenotypes, at least out to 5 years of age, an advantage of this model over existing transgenic and viral-mediated minipig and NHP HD models is that it was created using cDNA for full length human *mHTT* (67 exons), providing a full landscape for HTT-lowering therapeutic constructs (*Baxa et al., 2013*). Transgenic HD minipigs have a long prodromal period, show evidence of mHTT-positive inclusions in striatum and cortex, reduced DARPP-32 expression, microgliosis and mild WM demyelination beginning at 2 years of age (*Vidinská et al., 2018*). At 4–6 years of age, HD minipigs show more profound neuropathological changes and only at 6–8 years of age do they develop a mild motor phenotype, including impaired gait and reduced treat retrieval by tongue. More recently, a knock-in minipig model has been created by Exemplar Genetics that expresses full-length mHTT-150Q in all tissues evaluated, and piglets show uncoordinated hindlimb movement (unpublished data). The sheep and pig models have been employed in preclinical HD research to screen promising HTT-lowering therapeutics and one of these approaches, AAV-mediated delivery of a mHTT-specific microRNA, has recently advanced to the clinical trial stage (AMT-130), illustrating how critical these large animal models are to the field (*Vallès et al., 2021*; *Evers et al., 2018*).

In comparison to the transgenic HD sheep and minipigs, the HD NHPs reported here have a much shorter premanifest period and display many of the cardinal features of HD that are often used as primary and secondary outcome measures in HD human clinical trials, including motor and cognitive decline measured via the UHDRS and TFC (via the NHP clinical rating scale and SDR task in this model)

and regional brain atrophy that is quantified using MRI/DTI. Importantly, the behavioral and imaging findings in HTT85Q macaques correlated strongly with each other and constitute a comprehensive set of outcome measures that are be able to be used in future therapeutic evaluations in this model.

An advantage of viral-mediated HD macaques over the transgenic macaque models that have been created to date is that they can be generated in large enough numbers to appropriately power studies investigating new biomarkers of disease progression and promising therapeutics. While the transgenic HD macaque model (exons 1–10 of the *hHTT* gene with 67–73Q) created by Chan et al. showed several key disease symptoms (progressive motor phenotypes, evidence of impulsivity, and changes in temperament), along with WM microstructural changes, striatal atrophy, and aggregate formation, they only generated and characterized a small cohort of three transgenic animals, along with controls, precluding evaluation of any therapeutics (*Chan et al., 2014*; *Chan et al., 2015*; *Raper et al., 2016*; *Meng et al., 2017*; *Lallani et al., 2019*). In addition to being able to generate larger cohorts of animals, viral-vector-based macaque models allow for leeway in model design including the ability to assess different promoters, HTT fragment lengths, CAG repeat lengths, capsid serotypes (and combinations thereof) in a shorter timeframe, and with less expense, compared to transgenic and knock-in NHP models. For example, if desired, a longer prodromal phase in this AAV-based macaque model may be achievable by expanding the HTT fragment length (i.e., expressing the first 10 exons of human *mHTT* cDNA vs. the first 3 exons). Increasing the *mHTT* fragment length would also expand the real estate available for *mHTT*- lowering therapeutics. A longer prodromal phase may also be achieved by lowering the CAG repeat number, particularly given that HD patients show a direct correlation between CAG repeat length and age at disease onset (*Lee et al., 2012*). As the CAA/CAG cassette has been recently shown to associate with disease progression scores in human HD patients (*Ciosi et al., 2019*), removal or duplication of this cassette may be another mechanism by which the rate of disease progression can be hastened or prolonged, respectively. Another possibility is to engineer patient-relevant single nucleotide polymorphisms into the mHTT transgene for evaluation of allele-specific HTT-lowering therapeutics.

While the AAV2:2retro-HTT85Q model presented here offers advantages over some of the existing small and large HD animal models, there are limitations to acknowledge as well. Current viral vector packaging capacity limitations preclude expression of the full-length *HTT* gene and the N171 N-terminal fragment of mHTT is expressed instead. This size restriction extends to the control construct as well, and may be responsible for some of the intermediate behavioral effects that we observed in the 10Q group, as wildtype HTT is not typically cleaved by cellular proteases into smaller N-terminal fragments. Other limitations of this model include expression of HTT from a strong promoter (CAG) versus an endogenous *HTT* promoter and the fact that HTT85Q macaques have two wildtype, endogenous *HTT* alleles, in addition to the pathogenic human *mHTT* transgene. The HTT85Q transgene lacks introns, which precludes the formation of alternatively spliced variants (e.g., *HTTexon1*), known to contribute to mHTT-mediated toxicity (*Sathasivam et al., 2013*). Future optimization of the transgene design to include intron 1 may be able to get around this limitation. Finally, expression of mHTT in viral-based models is limited to areas of transduction, while HTT is expressed throughout the brain and in peripheral tissues. The current model employed a mixture of AAV2 and AAV2.retro delivered into the caudate and putamen, which achieved a much broader expression of mHTT throughout the brain compared to previous vector-based models, but expression is not ubiquitous.

As some of the HTT-lowering therapeutics currently under preclinical investigation are delivered using viral vectors (i.e., miRNAs, zinc finger protein-repressors, CRISPR-mediated gene editing, etc.) it will be critical to investigate the potential immune considerations of AAV-based therapeutics in this model, where neutralizing antibodies generated from creating the model could potentially reduce the efficacy of a second AAV vector to deliver its gene cargo. Should this be an issue, plasmapheresis, immunosuppression and/or capsid serotype switching using AAVs from distinct clades may be viable options. Other therapeutics also amendable to testing in this model include antisense oligonucleotides, stem cell-based therapeutics, small molecules, neurotransmitter-modulating pharmacotherapeutics, among others. Taken together, we are hopeful that the AAV2:AAV2retro macaque model will become a new resource for the HD research community, both for identifying novel biomarkers of disease progression as well as testing promising therapeutic candidates.

## Methods
### Experimental model and subject details
Animals

This study included 17 adult *Rhesus Macaques (Macaca mulatta)* (age 6–13; $n = 12$ female, $n = 5$ male) (*Table 1*). Monkeys were pair housed on a 12-hr light/dark cycle, provided with monkey chow rations twice daily, and given ad libitum access to water. Animal weights were recorded monthly by veterinary staff and heath checks were conducted by Oregon National Primate Research Center (ONPRC) veterinary and technical staff daily. The Institutional Animal Care and Use Committee (protocol: IP00000408) and the Institutional Biosafety Committee (protocol: 09-13) at the ONPRC and Oregon Health and Science University (OHSU) approved all experimental procedures, and all of the guidelines specified in the National Institutes of Health Guide for the Care and Use of Laboratory Animals (*National Research Council, 2011*) were strictly followed.

### Viral vectors
pAAV2.retro capsid plasmids were provided by the Karpova lab at the Howard Hughes Medical Institute (HHMI), Janelia Research Campus and pAAV2 capsid plasmids were supplied by the OHSU/ONPRC Molecular Virology Support Core. Plasmids containing the N171 N-terminal fragment sequence of human *HTT* were manufactured by GenScript and subsequently cloned into a transgene cassette flanked by viral inverted terminal repeats. The glutamine encoded repeat for HTT85Q contained 83 pure CAG repeats, followed by a single CAA/CAG cassette, while the glutamine encoded sequence for HTT10Q contained 8 pure CAG repeats followed by a single CAA/CAG cassette. Both constructs contained a proline stretch distal to the glutamine repeat in the following allelic conformation where $Q^T$ represents the total glutamine length:

HTT85Q: $Q^T = 85$, $(CAG)_{83}(CAACAG)_1(CCGCCA)_1(CCG)_7(CCT)_2$
HTT10Q: $Q^T = 10$, $(CAG)_8(CAACAG)_1(CCGCCA)_1(CCG)_7(CCT)_2$

*HTT* transgene expression was driven by a CAG promoter (cytomegalovirus [CMV] enhancer fused to the chicken beta-actin promoter) in both vector constructs. Recombinant AAV2 and AAV2.retro vectors were produced by the OHSU Molecular Virology Support Core and were prepared using a scalable transfection method as previously described in *Weiss et al., 2020*. Viral titers were determined by quantitative PCR of purified vector particles using a CAG primer/probe set: Forward: 5′-CCATCGCTGCACAAAATAATTAAAA-3′, Reverse: 5′-CCACGTTCTGCTTCACTCTC-3′, Probe: 5′-CCCCTCCCCACCCCCAATTTT-3′. Neutralizing antibody (Nab) assays were carried out as previously reported (*Weiss et al., 2020*), and animals with anti-AAV2 Nab titers of 1:20 or less were selected as study participants.

### Surgery
Pre-surgical procedures and surgical parameters were identical to those previously reported (*Weiss et al., 2020*), with a few noted deviations. Immediately prior to surgery, all animals received a 12.5-min, T1-weighted MRI scan, which was used for surgical targeting for each of the injection tracts using the Osirix Lite DICOM Viewer. Animals received four bilateral injections (eight injections total) of either a 1:1 mixture of AAV2 and AAV2.retro at a titer of 1e12 vg/ml or a buffered saline injection w/ F-Pluronic ($n = 6$ animals injected with AAV2-HTT85Q + AAV2.retro-HTT85Q, $n = 6$ with AAV2-HTT10Q + AAV2.retro-HTT10Q and $n = 5$ with phosphate buffered saline). Injections were made bilaterally into the pre-commissural caudate (90 µl) and putamen (95 µl) as well as the post-commissural caudate (60 µl) and putamen (85 µl), for a volume of 330 µl per hemisphere. Pre-commissural injection sites in the caudate and putamen were located, on average, 22–24 mm anterior to the earbar zeroing point (approximately 1 mm in front of the crossing of the anterior commissure), while post-commissural sites in these structures were located, on average, 18–19 mm anterior to the earbar zeroing point. Injection sites were spaced approximately 4–5 mm apart from one another in both the caudate and the putamen, determined by our previous work assessing the biodistribution of AAV2 and AAV2.retro in these brain structures (*Weiss et al., 2020*). Injections into the caudate were located, on average, 6 mm lateral to the sagittal sinus and 15–16 mm ventral to the pial surface. Injections into the putamen were located, on average, 12–14 mm lateral to the

sagittal sinus and 18–19 mm ventral to the pial surface. Infusate was administered using convection enhanced delivery, with the rate ramping up from 1 to 4 µl/min, with a 0.5 µl increase every 5 min. Following completion of all injections, dura was sutured closed, craniotomy sites were filled with gel foam, musculature and skin were sutured, and postoperative care was administered as previously described (*Weiss et al., 2020*).

## 3-Choice SDR task

The 3-Choice SDR assesses spatial working memory and was conducted in a Wisconsin General Testing Apparatus (WGTA) equipped with a plexiglass 3-well stimulus tray (6 cm in diameter, 2.5 cm depth, 8.5 cm apart; *Figure 1A*). Wells were arranged in a single row on the tray and were covered with identical plastic discs. At the beginning of each trial, an experimenter who was blind to the monkey's treatment group uncovered one of the wells, placed a preferred food reward in the well, and replaced the cover on the well. The screen was then lowered and a stopwatch started. After a delay, the screen was raised, and the monkey was allowed to displace one of the well covers. If the baited well was chosen, the monkey was allowed to retrieve the food reward, after which the experimenter lowered the screen and marked the trial correct. If the monkey uncovered one of the nonbaited wells, the screen was quickly lowered, the trial marked incorrect, and a new trial started. Prior to surgery, the animals were trained to complete this task to an 80% correct criterion using a 1 s delay. During the training sessions, positive reinforcement was employed, including the use of a clicker and a preferred food reward for correct choices. Negative reinforcement was never used. Fifteen of 17 animals successfully acquired the rules of the task (*Table 1*). At the baseline, 3-, 6-, 9-, 14-, 20-, and 30-month study timepoints, monkeys completed two sessions of this task, on two consecutive days, with variable delays of 1, 3, and 5 s (8 trials each), resulting in 48 trials total. Each response was recorded and an intertrial interval of 15 s was initiated before advancing to the next trial.

## Lifesaver Retrieval Task

Fine motor skill capability was assessed using the Lifesaver Retrieval Task. This task was conducted in a WGTA equipped with a plexiglass cage front containing two arm openings as well as a plexiglass stimulus tray with two vertical, metal posts (7.5 cm high, 11.5 cm apart) on the left and right side of the tray (*Figure 1D*). At the beginning of each trial, the experimenter threaded a Lifesaver candy onto one of the two posts. The screen was lifted to allow the monkey to retrieve the candy with their ipsilateral hand (right hand/right post, left hand/left post). For each session, the position of the Lifesaver alternated between the left and right post for 10 trails ($n = 5$ left, $n = 5$ right). Prior to surgery, all 17 animals successfully learned to complete this task (*Table 1*). During the training sessions, positive reinforcement was employed, including the use of a clicker and a preferred food reward (in addition to the lifesaver) for successful retrievals. Negative reinforcement was never used.Subjects completed two sessions of this task on two consecutive days at each timepoint (baseline, 3, 6, 9, 14, 20, and 30 months). The retrieval latency was recorded in seconds and defined as the time interval between when monkey's hand first passed through the fiberglass holes in the stimulus tray and ended when their hand was completely withdrawn back into the cage.

## Neurological ratings

An NHP-specific NRS was used to score motor behaviors. Using methods previously described (*McBride et al., 2011*), trained research staff who were blind to experimental treatment groups observed each monkey for 30–45 min in their home environment. Each behavioral phenotype on the NRS was scored between 0 and 3, with a score of 0 representing normal behavior and a score of 3 representing severely abnormal behavior (see *Supplementary file 3* for a list of each behavior scored). Monkeys were rated at baseline, prior to surgery, and at each month post-surgery, with the addition of a 2-week post-surgery timepoint. Ratings were conducted during the same time of day, between 12 PM and 4 PM. Additionally, we rated the animals using the NRS both prior to, and for 45 min following intra-muscular administration of the dopamine agonist, Apomorphine HCl (0.3 mg/kg, Sigma). Apomorphine ratings were conducted at baseline and 3-, 6-, 9-,14-, and 20-month post-surgery, but not at the 30-month timepoint.

## MRI acquisition

At 3-, 6-, 9-, 14-, 20-, and 30-month timepoints multimodal neuroimaging data were collected from the animals on study. All monkeys were anesthetized for the duration of the scanning session to prevent motion artifacts and to ensure their safety. Anesthesia was induced with ketamine HCl (15 mg/kg IM), and maintained via inhalation of 1–2% isoflurane gas vaporized in 100% oxygen. Animals were positioned on the scanner bed in a head-first supine orientation and immobilized in the head coil with foam padding. A fiducial marker (vitamin E tablet) was taped to the right side of the head prior to each scan. Blood oxygenation and heart rate were continually monitored throughout the scan by trained veterinary staff. Post scan, the animals were extubated, returned to their housing environment, and their recovery monitored closely for several hours by laboratory and veterinary staff.

A Siemens Prisma whole body 3T MRI system (Erlangen, Germany) with a 16-channel pediatric head rf coil was used to acquire all of the MR images. Four types of images were acquired: 3D T1-weighted magnetization-prepared rapid gradient-echo (MP-RAGE; *Mugler and Brookeman, 1990*), 3D T2-weighted sampling perfection with application optimized contrasts using different flip angle evolution (SPACE) (*Mugler et al., 2000*), DTI scans, and resting-state functional connectivity (rs-fMRI) scans. Details of the acquisition parameters for the structural (T1w/T2w) and DTI have been previously described and were identical to those reported by *Weiss et al., 2021*. Briefly, 3D SPACE imaging sequences were acquired with 0.5 mm isotropic voxels (TE/TR = 385/3200 ms, flip angle = 120°, 320 × 320 × 224). Three SPACE images were acquired in each session (total acquisition time 29 min 42 s). For 3D MP-RAGE imaging sequences, voxel sizes and the field of view were identical to the 3D SPACE images (TE/TR/TI = 3.44/2600/913 ms, flip angle = 8°). Similarly, three MP-RAGE images were acquired in each imaging session (total acquisition time 31 min 9 s). Diffusion-weighted (DW) volumes were acquired using a spin-echo planar imaging (EPI) sequence with 1.0 mm isotropic voxels and TR/TE = 6700/73 ms, GRAPPA factor = 2, echo train length = 52. Seven repetitions of 6 b0 volumes and 30 DW volumes with single $b$ = 1000 s/mm$^2$ and an anterior-to-posterior phase-encoding direction were acquired each session. To correct for susceptibility-induced distortions, a single b0 volume with a reversed (posterior-to-anterior) phase-encoding direction was also acquired (acquisition time 29 min 39 s). For rs-fMRI, BOLD images were acquired using T2-weighted GE-EPI sequence (TE/TR = 25/2290 ms and flip angle = 79°). Voxel sizes were 1.5 isotropic, and 784 3D volume images were collected. To control for effects of anesthesia on the BOLD signal, each rs-fMRI run started 45–50 min after the subject was first anesthetized with ketamine, and all animals were maintained on a constant level of 1% isoflurane during rs-fMRI acquisition. Following the rs-fMRI scan, there was a short reverse phase-encoded rs-fMRI scan (with 20 volume images) that was acquired for distortion correction (acquisition time 31 min 40 s).

## SPACE/MP-RAGE processing

SPACE and MP-RAGE images were processed using identical procedures. First, each of the three images collected during the imaging session were averaged. To accomplish this, the first scanned image from each modality was selected as the reference, and the other two images were registered to the reference with rigid-body transformations using ANTS (version 2.1, http://stnava.github.io/ANTs/; *Cook, 2022*) These warped images were subsequently averaged using FSL to produce merged T1w and T2w whole-head images (version 5.0, http://fsl.fmrib.ox.ac.uk/fsl/fslwiki/). Due to similarities in the image contrasts of the T2w, b0, and EPI sequences, registration between the imaging modalities is superior when T2w data are used for structural alignment, compared to T1w data (*Weiss et al., 2021*; *Adluru et al., 2012*). Additionally, data from human clinical populations have also demonstrated that T2w images provide improved sensitivity for VMB/TBM (*Diaz-de-Grenu et al., 2011*). Therefore, we used the ONPRC18 T2w template as our primary anatomical reference. For brain extraction, the merged T2w image was registered to the ONPRC18 T2w whole-head template (*Weiss et al., 2021*) with $b$-spline nonlinear transformations implemented in ANTS/FSL. Using the resulting registration parameters, the template brain mask was inversely mapped to the T2w merged images, verified by trained observers, and the brain extracted using FSL. Next, intensity bias correction was performed using the 'N4BiasFieldCorrection' tool in ANTS (*Tustison et al., 2010*). Finally, to coalign the structural scans, the merged T1w image was registered to the merged T2w image with a rigid-body transformation and skull stripped using the same mask. The merged T2w images were next aligned to ONPRC18 template space using $b$-spline nonlinear transformations in ANTs/FSL, and the resulting

Inverse Warp fields (representing the deformation from individual space to ONPRC18 template space) were used to calculate log Jacobian Determinant maps with the ANTs software package. Here, positive values denote areas where individual space was dilated to align with ONPRC18 space (and was therefore smaller), and negative values denote areas where individual space was contracted to align with ONPRC18 space (and was therefore larger). The log Jacobian Determinant maps from each subject at each timepoint were then analyzed using voxel-wise statistical comparisons implemented in FSL-SwE, described below.

### DTI processing

First, a susceptibility-induced off-resonance field (h) was calculated from the six pairs of b0 volumes with opposite phase-encoding direction using 'topup', included in the FSL library (https://fsl.fmrib.ox. ac.uk/) (*Andersson et al., 2003*). A brain mask was manually generated for the resulting unwarped b0 image, and applied to the DW volumes in a denoising step that was implemented in MATLAB (script provided by Dr. Sune Jespersen, Aarhus University; *Veraart et al., 2016*). To account for motion, the eddy current induced off-resonance field (e) and rigid-body transformations (r) between DW volumes were estimated simultaneously using 'eddy' (FSL). Then, the three transformations (h, e, and r) were combined into one warp field to correct the denoised DW volumes (*Andersson and Sotiropoulos, 2016*). Finally, the DTI-TK toolkit was used to fit the denoised, eddy corrected b0s and DW volume to a single tensor (DTI) model. The DTI tensor maps for each animal at each timepoint were next aligned to the ONPRC18 tensor template with *b*-spline nonlinear registrations, and parameter maps for FA, AD, RD, and MD were subsequently generated in template space using DTI-TK and then compared using voxel-wise statistical comparisons implemented in FSL-SwE, described below.

### Resting-state fMRI

Preprocessing rs-fMRI data were implemented with scripts from the Analysis of Functional Neuro Images (AFNI) Software package (https://afni.nimh.nih.gov) that were modified in-house for use with NHP data (*Grant et al., 2022*). The T2w SPACE brain images were used to co-register the resting-state data in anatomical space and transformed to ONPRC18 template space using the deformation fields generated by ANTs in the TBM pipeline described above. After regressing for signals of cerebral spinal fluid (CSF)and WM, motion, and outliers' censors (*Jo et al., 2010*), FSL-MELODIC software (https://fsl.fmrib.ox.ac.uk/fsl/fslwiki/MELODIC) was applied to the baseline data to derive group-level connectivity networks using ICA (*Beckmann et al., 2005*). This approach identified four independent components (ICs) which appeared to align with previously described resting-state networks identified in macaques (*Yacoub et al., 2020*; *Hutchison et al., 2011*) and humans (*Beckmann et al., 2005*; *Finn et al., 2015*; *Allen et al., 2014*; *Smith et al., 2009*), overlapped primarily gray matter regions, and possessed low-frequency spectral power (*Griffanti et al., 2017*; *Figure 4—figure supplement 1*). Based on these criteria, the remaining four ICs were not further examined (*Griffanti et al., 2017*; *Figure 4—figure supplement 2* ). To facilitate longitudinal group-level comparisons, DR analysis was applied to each timepoint using FSL tools, focusing on the four ICs of interest. *z*-Scores from each of these four group-level ICs identified at baseline were then regressed against the individual 4D datasets at all of the timepoints to produce variance normalized time-courses for each of the four ICs for each induvial subject, and *z*-score maps indicating subject-level covariance patterns with each of the four ICs brain-wide at each timepoint. These methods closely recapitulate the approaches used by *Grant et al., 2022*. Voxel-wise differences between groups were then analyzed using FSL-SwE, described below.

## Quantification and statistical analysis

### Analysis of behavioral data

To account for individual differences in task performance pre-surgery, all behavioral measures (% correct, retrieval latency, NRS score) were analyzed as change scores, calculated by subtracting the baseline values from each subsequent timepoint. Line charts and bar graphs were created using PRISM and IBM SPSS software packages illustrating group means (± SEM). Repeated measure ANOVAs, implemented in SPSS, were used to compare scores between groups over time. For the 3-Choice SDR task, delay was included as a third factor in the ANOVA; for the Lifesaver Retrieval Task, hand was included as a third factor; and for the NRS, apomorphine was included as a third factor. Significance

was defined by p < 0.05, and post hoc tests with the Bonferroni correction for multiple comparisons were used to compare significant main effects. Additional planned comparisons were conducted using independent-sample t-test comparing between groups at each timepoint separately, to further elucidate the time-course of any group differences. One animal from the 85Q group was unavailable for behavioral analyses at the 30 M timepoint (n = 5 vs. n = 6). All of the behavioral assessments were conducted by research staff that were blind to the experimental group of the animal. Additionally, all of the data were deidentified prior to statistical analysis using generic subject codes in place of animal ID numbers.

## Analysis of DTI, TBM, and rs-fMRI data

For all imaging modalities (TBM, DTI, and rs-fMRI), parameter maps (log Jacobian Determinants, FA, AD, MD, RD, and z-scores) were compared across timepoints using the FSL Sandwich Estimator (FSL-SwE) tool (*Guillaume et al., 2014*) with threshold free cluster enhancement, 500 permutations (*Winkler et al., 2014*), and a gray matter mask, a WM mask, or a whole brain mask derived from the ONPRC18 labelmap. A threshold of p < 0.01 was applied to all the results, however, due to the small sample sizes in this study, family-wise error correction was not applied. Visualizations of the thresholded p-value maps were created using FSLeyes. To summarize the regional distribution of significant voxel-wise changes, for each contrast we calculated the percent of p < 0.01 voxels in each ROI in the ONPRC18 labelmap. This was accomplished by first using FSL tools to threshold and binarize and the p-value maps at p < 0.01 and then AFNI tools (3dcalc) to extract volume information. Histograms were subsequently plotted to illustrate the percent volume of each ROI with significant changes from baseline. Finally, to assess the magnitude of the changes, 'difference' maps were calculated for each imaging parameter (log Jacobian Determinants, FA, AD, MD, RD, and z-scores) by subtracting baseline from each subsequent timepoint (implemented using fslmaths commands). Then, for each parameter, masks were created from the thresholded p-value maps at each timepoint, and the average 'difference' value calculated under each p-value mask. This calculation was made brain-wide, rather than for each ROI separately. Using these data, the changes in the neuroimaging parameters were assessed with repeated measure ANOVAs including post hoc group comparisons using the Bonferroni correction; additionally, the behavioral and imaging measures were collapsed across time (averaged) and correlated using one-tailed Pearson correlations in SPSS. One animal from the 85Q group was unavailable for imaging analyses at the 30 M timepoint (n = 5 vs. n = 6). All of the imaging measurements were collected by research staff that were blind to the experimental group of the animal. Additionally, all of the imaging data were deidentified prior to statistical analysis using generic subject codes in place of animal ID numbers.

## Tissue collection

Necropsies were performed as previously reported in *Weiss et al., 2020*. Brains were cut into 4-mm-thick slabs in a rhesus brain matrix (ASI Instruments, MBM-2000C). Slabs from the left hemisphere were used for sub-regional tissue collection; tissue punches were snap frozen, stored at −80°C and are dedicated to future molecular studies. Slabs from the right hemisphere were post-fixed at 4 degrees Celsius in 4% paraformaldehyde (PFA) for 48 hr and then placed into a 30% sucrose solution at 4 degrees Celsius for 2 weeks (30% sucrose solution was changed out every 4 days for fresh solution). Slabs were then sectioned coronally into 40-μM-thick sections on a freezing microtome and tissue sections were stored at 4 degrees Celsius in a cryoprotective solution containing 30% sucrose and 30% ethylene glycol in phosphate buffered saline. Sodium azide (0.1%) was added to the cryproctective solution as an antimicrobial agent. For longer term storage, tissue sections were stored at -20 degrees Celsius in cryoprotective solution. Needle tracts corresponding to the surgical injection sites were verified in animals from all groups during tissue collection and cutting.

## Tissue staining and microscopy

### Immunohistochemistry

Tissue sections from all animals were rinsed in Tris-buffered saline (TBS) with 0.05% Triton X-100 before being blocked in TBS solution containing 10% normal goat serum (Gibco). Sections were incubated in an em48 (Millipore, MAB 5374, 1:500) or 1–82 aa/2B4 (MAB 5392, 1:500) primary antibody solution overnight. Sections were washed and incubated in a goat anti-mouse biotinylated secondary

antibody solution for 1 hr (Vector Labs, BA-9200, 1:500). Tissue was rinsed and the signal was developed using a standard Vectastain ABC kit (Vector Laboratories, PK6100) for 1 hr before developed in 3,3′-diaminobenzidene (DAB) with nickel (II) sulfate (0.05% DAB, 2.5% nickel (II) sulfate hexahydrate, 0.02% $H_2O_2$, Tris-buffered saline).

## Triple-label immunofluorescence

Native tissue autofluorescence was quenched by under a 4500 lumen, broad spectrum LED light for 48 hr (tissue sections in dishes containing PBS). Tissue was then rinsed and blocked as reported above and incubated in a primary antibody solution containing either em48 (1:250) and NeuN (Millipore, ABN78, 1:500) or em48 and GFAP (Dako, 20334, 1:1000). Secondary antibody incubation was the same for either condition, 1.5 hr in solution with Alexafluor Goat anti-Mouse 488 (Invitrogen, 1:500) for em48 visualization followed by 1 hr in Goat anti Rabbit 546 (Invitrogen, 1:500) to visualize NeuN or GFAP. Nuclei were labeled with a 30-s incubation in Hoechst 33342 (Invitrogen, H3570, 1:10,000).

## Microscopy

Brightfield images were captured on an Olympus BX51 microscope using 4X (U PlanS Apo, NA 0.16) and 20X (UPlanFL N, NA0.50) objectivesand an Olympus DP72 camera. ×20 images were captured as extended field images using CellSens software (version 1.15) Immunofluorescence images were captured on a Leica SP5 confocal microscope using a ×63 oil immersion objective (HCX Plan Apo lambda CS, NA 1.4), scan speed 1024 x 1024. Images displayed in *Figure 5B, C* were captured from a single *z*-plane (vs. a *z*-projection), were cropped and magnified from the ×63 image and brightness and gain of each channel were optimized using LAS AF software (Leica Application Suite, Advanced Fluorescence Version 2.7.3.9723).

## Acknowledgements

We extend our sincere gratitude to the ONPRC Division of Animal Resources and Research Support for the excellent care provided to the rhesus macaques involved in this study, with special acknowledgement of the steadfast and expert veterinary efforts that Lauren Drew Martin, Theodore Hobbs, Melissa Berg, Brandy Dozier, Rob Zweig, Michael Reusz, Alona Kvitky, Kristy Ritchie, Amy Kujacznski, and Isabel Bernstein contributed to this work.

## Additional information

### Funding

| Funder | Grant reference number | Author |
| --- | --- | --- |
| National Institutes of Health | NS099136 | Jodi L McBride |
| National Institutes of Health | F32NS110149 | Alison R Weiss |
| National Institutes of Health | T32AG055378 | Alison R Weiss |
| National Institutes of Health | P51OD011092 | Jodi L McBride |
| The Bev Hartig Huntington's Disease Foundation | | Jodi L McBride |
| National Institutes of Health | S10RR024585 | Sathya Srinivasan |

The funders had no role in study design, data collection, and interpretation, or the decision to submit the work for publication.

## Author contributions

Alison R Weiss, Conceptualization, Data curation, Formal analysis, Investigation, Methodology, Supervision, Validation, Visualization, Writing – original draft, Writing – review and editing; William A Liguore, Kristin Brandon, Conceptualization, Data curation, Formal analysis, Supervision, Validation, Visualization, Writing – original draft, Writing – review and editing; Xiaojie Wang, Zheng Liu, Formal analysis, Supervision, Visualization, Writing – review and editing; Jacqueline S Domire, Supervision, Project administration; Dana Button, Supervision, Visualization, Writing – review and editing; Sathya Srinivasan, Methodology, Resources, Visualization; Christopher D Kroenke, Conceptualization, Data curation, Formal analysis, Investigation, Funding acquisition, Methodology, Supervision, Validation, Visualization, Writing – review and editing; Jodi L McBride, Conceptualization, Formal analysis, Investigation, Funding acquisition, Supervision, Validation, Visualization, Writing – original draft, Project administration, Writing – review and editing

## Author ORCIDs

Alison R Weiss http://orcid.org/0000-0002-1821-8193
Jodi L McBride http://orcid.org/0000-0002-4816-7569

## Ethics

The Institutional Animal Care and Use Committee (protocol: IP00000408) and the Institutional Biosafety Committee (protocol: 09-13) at the ONPRC and Oregon Health and Science University (OHSU) approved all experimental procedures, and all of the guidelines specified in the National Institutes of Health Guide for the Care and Use of Laboratory Animals were strictly followed.

## Decision letter and Author response

Decision letter https://doi.org/10.7554/eLife.77568.sa1
Author response https://doi.org/10.7554/eLife.77568.sa2

# Additional files

## Supplementary files

• Supplementary file 1. Planned group comparisons for 3-Choice Spatial Delayed Response (SDR) task using one-tailed independent sample *t*-tests at each timepoint. $*p < 0.05$, $**p < 0.01$, $***p < 0.001$.

• Supplementary file 2. Planned group comparisons for Lifesaver Retrieval Latencies using one-tailed independent sample *t*-tests at each timepoint. $*p < 0.05$, $**p < 0.01$, $***p < 0.001$.

• Supplementary file 3. Neurological rating scale (NRS) for nonhuman primates. Behaviors were scored cage-side by trained observers, blinded to treatment condition, during 30- to 45-min focal observations. Scores were summed across categories to generate a total NRS score. Higher scores indicate more severe phenotypes.

• Supplementary file 4. Planned group comparisons in monthly neurological rating scale (NRS) scores using one-tailed independent sample *t*-tests at each timepoint. $*p < 0.05$, $**p < 0.01$, $***p < 0.001$.

• Supplementary file 5. Planned comparisons for pre- versus post-apomorphine neurological rating scale (NRS) scores at each timepoint using one-tailed paired-sample *t*-tests for each group separately. $*p < 0.05$, $**p < 0.01$.

• Transparent reporting form

## Data availability

The rhesus macaque brain atlas used to assess neuroimaging data has been previously published and uploaded to the Neuroimaging Tools and Resources Collaboratory (NITRC) https://www.nitrc.org/projects/onprc18_atlas. Statistical results reported in Supplementary Files 1, 2, 4, 5.

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
