## [Editor Report]

The authors show the utility of an AAV-based approach in non-human primates to develop an improved model of Huntington's disease. They have presented a convincing, carefully executed, body of work that will be of benefit to a range of researchers studying HD or developing therapies for HD. While this extends the work from an earlier paper (that presented the tools used to induce phenotypes) the results presented are new, relevant, and important to the community.

---

## [Decision Letter]

**Decision letter after peer review:**

Thank you for submitting your article "A novel rhesus macaque model of Huntington's disease recapitulates key neuropathological changes along with progressive motor and cognitive decline" for consideration by *eLife*. Your article has been reviewed by 2 peer reviewers, and the evaluation has been overseen by a Reviewing Editor and Jeannie Chin as the Senior Editor. The reviewers have opted to remain anonymous.

Essential revisions:

In addition to the detailed comments in the individual reviews below, the following are essential revision requirements, including new data, that should be addressed before resubmitting your manuscript.

1) At least one additional behavioral data point, at least 3-6 months beyond current data points, needs to be presented. This data is necessary to be able to assess the "progressive" nature of the disease as described in the manuscript.

2) Neuropathology from at least one animal in which distal changes in MRI are seen. If distal neuropathology is observed, this would suffice to demonstrate the potential of this model.

*Reviewer #1 (Recommendations for the authors):*

1. It is not clear from the manuscript if the rating and behavioural assessments were conducted blind or double blind to experimental group, or who it was that assessed the performance etc. More details of the testing and control of data analysis would be useful.

2. There are no co-ordinates given for the injection sites, and no imaging data to show how comparable the lesion sites were between animals. It would be interesting to know how similar the placement of the lesions was.

3. Line 158 they say that the 85Q animals performed worse (sic) than the 10Q controls, but this did not reach significance. If it is not significant, then it cannot be 'worse'. Please remove this statement.

4. It was interesting to see that despite a large number of parameters being measured, the behavioural changes were small with surprisingly few robust deficits detected, considering the range of measures possible. It would be interesting to have the authors view on why so few deficits emerged.

5. There are no data showing brain pathology, suggesting that the animals are still alive and under investigation. Extending the behavioral component of the study by 12 months would answer some critical questions about whether or not the lesions are resolving, and which of the behaviours would persist and which would recover to baseline levels. This is certainly what happens in rats given acute lesions (e.g., of substantia nigra or neostriatum).

6. It is not clear how much neurotoxicity actually occurred in the lesion sites. Recent advances in imaging have shown that PET imaging detects aggregate pathology. Given the absence of any neuropathology data, leading to my assumption that these animals are still alive, it would be interesting to know if the authors could detect aggregates in either caudate/putamen where AAVs were injected (which would be expected) or in the cortex where secondary pathology was detected. This would make the study very interesting and would improve the potential usefulness of the model.

*Reviewer #2 (Recommendations for the authors):*

Page 3

Line 40 – the authors refer to glutamine stretch. We now know that what is relevant is the CAG stretch (citations required)

Lines 62-63 – cite the major therapeutic advances made with prior models (ASO, miRNAs). While old, they led to clinical trials that are either halted (ASO) or in process (miRNA). The latter is actually the senior authors' work? How did rodent models mislead clinical translatability or outcomes? With regards to neuropathologies. rodent models revealed the somatic expansion possibilities but as noted above they require genetic constructs, not transgenes.

Page 7

Lines 46-48 – the sentence structure could be improved for clarity. As it reads it is very confusing.

Why do Buffer and 10Q animals differ in so many of the assays? Additional discussion is warranted.

The discussion recapitulates in great detail many of the results and can be significantly shorter. The caveats discussed above should be brought to the readers' attention. It would be interesting to speculate whether this approach (with a different construct) could be used to assess the somatic expansion in NHPs (not just disease duration).

Figure 2 is extremely confusing and the legend could use strengthening. Please define the concentric circles, the numbers in the circles (which are too small to read), and the red shapes. It’s evident that the shapes get bigger and the lines approach the outer circles, but clarity on their meaning would help the general reader understand the importance of these measurements in defining the disease progression.

With Figure 3B, is there a way to show only a portion of the data (so that it is readable) and move some to the supplement? Also, the FA data is difficult to discern. The legend refers to red and blue voxels, but only purple is noted. Figure 4 has similar issues, but the abbreviations help. Maybe for 3B, 4, 5 and 6 only early and late timepoints would be necessary in a main figure to convey the major points and then put a complete time course in the supplement.

---

## [Author Response]

Essential revisions:Reviewer #1 (Recommendations for the authors):1. It is not clear from the manuscript if the rating and behavioural assessments were conducted blind or double blind to experimental group, or who it was that assessed the performance etc. More details of the testing and control of data analysis would be useful.

All of the behavioral assessments were conducted by research staff that were blind to the experimental group of the animal. All of the neuroimaging data were collected by research and veterinary staff that were blind to the experimental groups as well. Additionally, all of the data were de-identified prior to statistical analysis using generic subject codes in place of animal IDs. We thank the reviewers for this comment and have added these additional details to the manuscript as suggested (lines 172, 733-735, 756-759).

2. There are no co-ordinates given for the injection sites, and no imaging data to show how comparable the lesion sites were between animals. It would be interesting to know how similar the placement of the lesions was.

AAV injection coordinates were determined using MRI scans that were collected immediately prior to the surgery. Exact coordinates deviated slightly across animals based on the size and shape of the caudate and putamen in each animal, as well as the location of various intra-sulcal blood vessels that were avoided while planning trajectories to the target. Pre-commissural injections were planned approximately 1 mm in front of the anterior commissure and post-commissural injections were planned 4-5 mm posterior to the pre-commissural sites. Distance between injection sites was determined by our previous work assessing the biodistribution of AAV2 and AAV2.retro in these brain structures (Weiss et al., 2020). Anterior-posterior, medial-lateral and dorsal-ventral coordinate values were averaged across all animals and have been included in the Surgery sections of the Methods (Lines 566-582). Needle tracts corresponding to the planned injection sites were verified in all animals during tissue collection and cutting, as well as in em48 and 2B4-stained tissue sections. Overt lesions in the caudate and putamen were not observed in these animals; rather, widespread expression of mHTT aggregates were detected via IHC, now included in the Results section (Lines 350-379) and in Figure 5. The expression of mHTT in 85Q-treated animals lead to atrophy of specific brain structures, including the caudate and putamen, as well as distal gray matter regions as detailed in Figure 3.

3. Line 158 they say that the 85Q animals performed worse (sic) than the 10Q controls, but this did not reach significance. If it is not significant, then it cannot be 'worse'. Please remove this statement.

For the working memory task, group differences between 85Q and 10Q reached p<0.05 levels of significance starting at the 6-month timepoint, and persisting through 30-months (see Supplementary File 1). The Results section was clarified to improve this point, and the term “worse” was removed as requested. It now reads:

“Significant group differences also emerged between 85Q and 10Q groups at the 6-month timepoint and persisted through 30-months, with 85Q animals showing deficits in working memory performance that were not exhibited by either control group.” (Lines 137-139).

4. It was interesting to see that despite a large number of parameters being measured, the behavioural changes were small with surprisingly few robust deficits detected, considering the range of measures possible. It would be interesting to have the authors view on why so few deficits emerged.

Over the course of this 30-month (2.5 year) study, every single one of our 85Q-treated animals exhibited a range of motor phenotypes that have not been consistently reported in HD models using other species including: orofacial chorea, postural/gait/balance changes, altered treat retrieval and fine motor coordination, tremor, and bradykinesia. The severity of these phenotypes remained relatively mild over the course of the study, typically scoring 1 or 2 (out of 3) in each NRS category and reaching totals in the range of ~7, but the number of different phenotypes we observed increased over time. 85Q-treated animals also exhibited significant and consistent decreases in spatial working memory. Compared to their own pre-surgical baseline scores, performance in 85Q-treated animals fell by ~10%. This indicates a significant, but mild, deficit in spatial working memory. Taken together, we think these data illustrate that several consistent and measurable behavioral deficits were indeed detected in this model.

To clarify one point further, our intended goal was to create an animal model that would exhibit clear and reproducible motor and cognitive deficits, but we did not want to generate animals with phenotypes so severe as to interfere with activities of daily living (eg. ability to eat/drink on their own, or move around without risk of injury from falling). In terms of the range of possible scores commented on by the Reviewer, the NHP Neurological Rating Scale (NRS) used in this study (Supplementary File 3) was designed to be applicable to a number of different disease models that we work on, and so there are components on it that we did not expect to see in this model (eg. ataxia, dysmetria). Furthermore, in our experiences working on a severe NHP model of CLN7 Batten Disease, animals with NRS scores above ~25 points become increasingly challenging to care for appropriately, and often require specialized housing and clinical interventions that can confound rigorous and controlled comparisons across treatment groups. For this reason, we were satisfied with the level and severity of the behavioral deficits we observed in the 85Q-treated animals, and think they are most appropriate to model the prodromal and early-stages of HD.

5. There are no data showing brain pathology, suggesting that the animals are still alive and under investigation. Extending the behavioral component of the study by 12 months would answer some critical questions about whether or not the lesions are resolving, and which of the behaviours would persist and which would recover to baseline levels. This is certainly what happens in rats given acute lesions (e.g., of substantia nigra or neostriatum).

We extended the timeline for this study to 30-months and re-analyzed all of the data with this additional timepoint included. 85Q animals showed no evidence of functional recovery between the 20- and 30-months timepoints. Working memory deficits and fine motor skills did not improve between 20- and 30-months (Figure 1C and 1F). Additionally, neurological rating scores remained in the 5-7 range between the 20- and 30-month timepoint and did not show any sign of returning towards baseline levels. Taken together, we feel confident that these results do not show evidence of any significant functional recovery, out to 2.5 years (30-months).

6. It is not clear how much neurotoxicity actually occurred in the lesion sites. Recent advances in imaging have shown that PET imaging detects aggregate pathology. Given the absence of any neuropathology data, leading to my assumption that these animals are still alive, it would be interesting to know if the authors could detect aggregates in either caudate/putamen where AAVs were injected (which would be expected) or in the cortex where secondary pathology was detected. This would make the study very interesting and would improve the potential usefulness of the model.

We respectfully disagree with the Reviewer that the previous version of the manuscript was uninformative on this topic. Histological studies are indeed a “gold-standard”, but the knowledge about HD brain pathology that has been garnered from neuroimaging studies in patients is undeniable (e.g., TRACK-HD, and others). The primary goal of this paper was to report on the longitudinal in-vivo characteristics of this model. For this reason, we wanted to use well-established neuroimaging techniques that have been employed extensively in human studies—both in the field of HD and beyond.

We also wanted to directly address the Reviewer’s question regarding aggregate pathology. Animals were taken to necropsy after the 30-month timepoint and their brain tissues were collected for histological study. We added a figure to the manuscript showing representative IHC and IF staining of em48+ mHTT aggregates in many different brain regions, including the caudate and putamen (injected regions), as well as in several other cortical and subcortical structures that project to these two regions. The results (illustrated in Figure 5) demonstrate widespread and robust mHTT-mediated expression in the brains of the 85Q monkeys in the same regions where significant pathology was detected via neuroimaging. A more detailed histological characterization is planned to assess gray matter cell loss and axonal/myelin integrity in this model. Molecular studies will also investigate potential somatic instability across brain regions. However, this work is beyond the scope of this manuscript, and we plan to report these data separately.

In addition to the post-mortem validation provided here in this revised manuscript (see Figure 5), we have conducted two additional studies that address this point. Between 16-20 months after surgery, we quantified D2/3 receptor densities and cerebral glucose metabolism with F18-Fallypride and F18-FDG PET, respectively. We recently submitted the results from this study and the manuscript is currently under review.

Furthermore, just prior to necropsy at 30 months post-surgery, we also quantified mHTT aggregate burden in this model in vivo using a novel PET tracer that binds to aggregated forms of mHTT. This tracer was recently validated in HD mice (Bertoglio et al., 2022). 85Q-treated animals showed significantly increased binding of this ligand in areas of the basal ganglia (including in the head of the caudate and throughout the putamen), but also in distal cortical areas like the prefrontal, premotor, supplemental motor and anterior cingulate cortices (now mentioned in the discussion, lines 440-446). Critically, this work in our NHP model revealed the formation of radio-metabolite species that were not previously observed in mouse studies. Moreover, the serial blood sampling that is necessary for appropriate metabolite correction and creation of an arterial input function was not possible in the HD mouse models due to limited blood volume. Therefore, protocols and methodology that we developed in this NHP model allowed for advancement of this putative imaging biomarker using similar techniques to those used in the clinical investigation of this ligand in human HD patients. Moreover, the results that we obtained using both VOI- and voxel-based analyses will help to inform analyses on the ongoing human clinical data set. This work is being submitted in a separate manuscript.

Reviewer #2 (Recommendations for the authors):Page 3Line 40 – the authors refer to glutamine stretch. We now know that what is relevant is the CAG stretch (citations required)

These sentences in the Introduction now read:

“Huntington’s disease (HD) is a genetic, progressive neurodegenerative disorder caused by an expanded CAG/CAA repeat in exon 1 of the *HTT* gene. When the CAG stretch exceeds approximately 40 repeats, the encoded HTT protein misfolds and sets off a toxic sequence of events inside the cell including transcriptional dysregulation, mitochondrial dysfunction, calcium signaling disruption and altered neurotransmission, including glutamatergic and dopaminergic dysregulation.” (Line 39-43).

Lines 62-63 – cite the major therapeutic advances made with prior models (ASO, miRNAs). While old, they led to clinical trials that are either halted (ASO) or in process (miRNA). The latter is actually the senior authors' work? How did rodent models mislead clinical translatability or outcomes? With regards to neuropathologies. rodent models revealed the somatic expansion possibilities but as noted above they require genetic constructs, not transgenes.

The therapeutic advances made with prior models are included in the discussion (lines 475-478) when comparing the utility of the different large animal models, including the work that led to the ASO and AAV-miRNA based clinical trials. The preclinical work that led to the AAV5-miRNA clinical trial was conducted by UniQure in HD mice, HD transgenic pigs and a smaller number of naïve macaques, which is mentioned in the discussion. Rodent models have not misled clinical translatability, per say. Rodent models do have specific limitations regarding brain size, neuroanatomy, behavior, etc which is detailed in the introduction in lines 60-73 as well as the discussion in lines 454-459. While some of the rodent models have revealed somatic expansion, and this is one advantage of these specific models, the other caveats mentioned above make it difficult to use them alone, without the use of a large animal model, in addition. It is the authors’ opinion that using a combination of animal models that appropriately display different aspects of human HD is likely the best strategy for evaluating therapies.

Page 7Lines 46-48 – the sentence structure could be improved for clarity. As it reads it is very confusing.

We edited the sentence for grammar and clarity. Please refer to Line 47-50.

Why do Buffer and 10Q animals differ in so many of the assays? Additional discussion is warranted.

We appreciate this comment and agree that the data points to an intermediate behavioral effect at some timepoints in 10Q-treated animals on the Lifesaver and Spatial Delayed Response tasks. Animals in the 10Q group do not show impairments on these tasks, however, as the 85Q-treated animals do. Rather, they improve on each task over time, but not to the same degree as the buffer-injected controls. However, some of the Buffer and 10Q animals did exhibit mild phenotypes, as rated on the NRS, at 0.5-, 21- and 28-months post-surgery (lines 183-184). It is noteworthy that none of these effects were apparent in the neuroimaging data. Regarding intermediate changes seen in the 10Q group, these findings suggest that overexpression of N-terminal fragments of HTT bearing a CAG repeat length of 10 may have some deleterious effects on cellular function, but does not induce overt structural changes. Our histological evaluations of the 10Q animals using 2B4 and EM48, as well as our mHTT-PET study, revealed no evidence of aggregate formation in 10Q. However, this does not preclude the presence of other cellular changes in the cell that could be impairing behavior. For this reason, designing the optimal HTT control construct will require continued refinement (added to the discussion, line 506-520).

The discussion recapitulates in great detail many of the results and can be significantly shorter. The caveats discussed above should be brought to the readers' attention. It would be interesting to speculate whether this approach (with a different construct) could be used to assess the somatic expansion in NHPs (not just disease duration).

We revised the majority of the discussion to address each of the Reviewers’ comments—including improving on clarity and brevity. Caveats regarding the model that were brought up by both reviewers have been addressed in the discussion.

Figure 2 is extremely confusing and the legend could use strengthening. Please define the concentric circles, the numbers in the circles (which are too small to read), and the red shapes. It’s evident that the shapes get bigger and the lines approach the outer circles, but clarity on their meaning would help the general reader understand the importance of these measurements in defining the disease progression.

We agree with Reviewer 2 that Figure 2 was confusing. This figure has been removed from the revision, which was there originally to provide further detail on the observed behavioral phenotypes reported in Figure 1G and 1H. The text in the results and Discussion sections still provides details about the specific behavioral phenotypes we observed in 85Q-treated animals, including the timeframes during which specific behaviors emerged throughout the study (lines 185-195 and lines 405-409).

With Figure 3B, is there a way to show only a portion of the data (so that it is readable) and move some to the supplement? Also, the FA data is difficult to discern. The legend refers to red and blue voxels, but only purple is noted. Figure 4 has similar issues, but the abbreviations help. Maybe for 3B, 4, 5 and 6 only early and late timepoints would be necessary in a main figure to convey the major points and then put a complete time course in the supplement.

We replotted the MRI, DTI and RSFC imaging figures (Figures 2-4) to improve readability and clarity, as well as to include data out to the 30-month timepoint (requested by Reviewer 1). To accomplish this, we made the following changes:

1. Given the lack of p<0.01significant voxels in the 10Q and Buffer groups at any timepoint, we replaced the top panel in all of the imaging figures (that previously included columns showing 85Q, 10Q and Buffer groups) with panels showing 9 different axial levels across all timepoints for 85Q-treated animals only (to provide a clearer and more thorough view of the results). Color maps were also revised to avoid purple and navy colors—decreases are now shown in blue/cyan and increases are shown in red/yellow. See figure 2A, 3A, and 4A.

2. We re-plotted the histograms in 2B, 3B and 4B and included abbreviations for the brain regions to enable the use of larger fonts that are easier to read.